# Trust the model when it is confounded: Model-based Reinforcement learning for Confounded POMDPs

## Abstract

We consider model-based reinforcement learning (MBRL) in confounded partially observable Markov decision processes (POMDPs), where unobserved confounders in the environment will introduce bias into the learned dynamics model. Existing studies either rely on a kernel function to model the environment, or tabular settings where the observation spaces are discrete. To address these limitations, we propose a deep proximal causal MBRL (DPC-MBRL) method. Specifically, we first establish a consistent identification result for the policy value in confounded POMDPs through proximal causal inference. Based on this identification result, we then employ neural networks to model the environment dynamics, which enables a more flexible function approximation than existing studies. Through experiments on an advanced physics simulation benchmark MuJoCo and a real-world medical dataset, we demonstrate that DPC-MBRL mitigates the bias induced by unobserved confounders and yields more accurate dynamics model estimates than standard MBRL approaches.

## 1 Introduction

Model-based reinforcement learning (MBRL) relies heavily on robust modeling of the true environment, and many studies adopt computationally expensive function approximators or model ensembles to improve model estimation accuracy (Nagabandi et al., 2018; Kurutach et al., 2018; Janner et al., 2019; Yu et al., 2020; Frauenknecht et al., 2024). However, in partially observable environments, the presence of unobserved confounders, which affects both actions and outcomes (next state and reward), makes it inherently challenging to achieve an accurate characterization of the environment's dynamics (Polydoros & Nalpantidis, 2017; Ding et al., 2023; Hwang et al., 2024; Lambrechts et al., 2024).

In this work, we are interested in scenarios where interaction data are collected from a partially observable Markov decision process (POMDP) (Singh et al., 2021; Kurniawati, 2022). For example, imagine a robot trying to learn to walk quickly and steadily in complex terrain. We wish the robot walk faster on flat ground as shown in Figure 1(a) and more slowly on steep ground as in Figure 1(b). However, because of the absence of the corresponding sensors or cameras, certain states may remain unobservable, such as a puddle of water on the ground. A naive robot might infer that it should walk slowly on flat ground to remain steady (Figure 1(c)) even if the ground is dry as shown in Figure 1(d), which is contrary to the desired actions in Figure 1(a). Such an unobserved state which affects both the robot's actions and the next state transitions or reward (i.e., Water as highlighted by the red dashed box in Figure 1) is known as an *unobserved confounder* in causal inference (Kuroki & Pearl, 2014; Tchetgen Tchetgen, 2014; Miao et al., 2018). A partially observable environment containing such confounders is also referred to as a *Confounded POMDP* in literature (Tennenholtz et al., 2020; Miao et al., 2022; Shi et al., 2022; 2024). Under this setting, naively applying off-the-shelf MBRL algorithms without accounting for the unobserved confounders can lead to erroneous or suboptimal policies (Nagabandi et al., 2018; Kurutach et al., 2018; Janner et al., 2019; Yu et al., 2020; Frauenknecht et al., 2024), such as walking slowly on a flat and dry ground in the robot example.

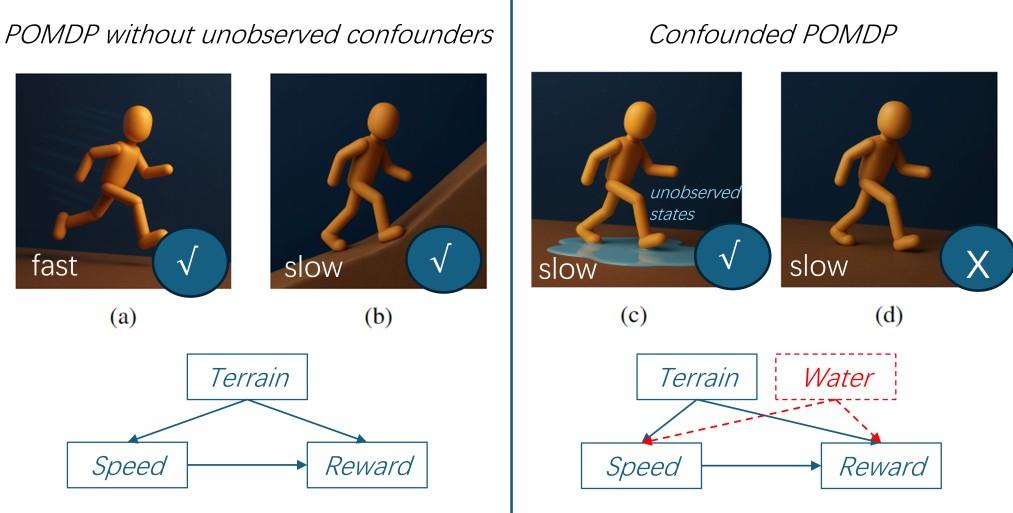

Figure 1: The left panel illustrates the robot's behavior in POMDP without unobserved confounders: (a) moving quickly on flat ground, and (b) walking slowly on a steep slope to maintain stability. The right panel illustrates the robot's action decisions in a confounded POMDP when the unobserved confounder is not dealt with: (c) without detecting the presence of water, the robot walks slowly on a slippery flat ground to avoid falling, and (d) it mistakenly walks slowly even on dry flat ground. Shown below are the causal relationships between the state, the action, and the reward. In the confounded POMDP, water is an unobserved confounder that influences both the robot's action decisions and the obtained reward.

To mitigate the bias introduced by unobserved confounders in the environment, two recent studies have leveraged causal inference techniques to develop MBRL algorithms for confounded POMDPs (Gasse et al., 2023; Hong et al., 2024). Nevertheless, both approaches exhibit major limitations that hinder their practical applications. Gasse et al. (2023) proposed a model-based approach for policy value identification in finite discrete environments, where the learned dynamics model relies on a tabular representation. Consequently, this approach does not scale to continuous state and action spaces. Hong et al. (2024) employed kernel functions to model the dynamics model, thereby proposing a nonparametric identification method capable of handling continuous observation spaces. However, this method relies on prespecified kernel functions, which incur high computational costs in high-dimensional settings and require precise parameter tuning (Indu & Dimri, 2023). Moreover, their work lacks empirical evaluation to demonstrate its practical effectiveness. We also note a series of other studies that have developed RL algorithms for confounded POMDPs, but in a *model-free* context (Tennenholtz et al., 2020; Shi et al., 2022; Miao et al., 2022; Lu et al., 2023; Bennett & Kallus, 2024). These model-free methods are grounded in the Bellman equation and do not explicitly model the environment dynamics, which are not applicable to a model-based context and are not discussed further here. For a more comprehensive discussion of related work, please refer to Appendix B.

Motivated by the aforementioned limitations, it is imperative to develop a more practical MBRL method capable of mitigating the confounding effects arising from unobserved confounders in the environment, without relying on prespecified kernels or tabular settings. Inspired by a recent study on proximal causal learning (Kompa et al., 2022) for deconfounding in the presence of latent confounders in causal effect estimation, we propose a deep proximal causal MBRL (DPC-MBRL) method for confounded POMDPs. Specifically, we establish a consistent identification result for estimating the policy value via a sequence of bridge functions with proxy variables in confounded POMDPs. This identification result guarantees an unbiased estimate of the target policy value using only observable data, eliminating the bias introduced by unobserved confounders. Furthermore, we employ neural networks to construct the bridge functions, which allows a general approximation

for the dynamics modeling without being restricted to discrete observation spaces or the need for a prespecified kernel function.

The primary contributions of our work are summarized as follows:

- **Theoretical guarantee:** We establish a consistent model-based identification result for the policy value in confounded POMDPs through proximal causal inference and provide theoretical guarantees.
- **Deconfounding algorithm** We propose a deep proximal causal MBRL (DPC-MBRL) algorithm that achieves unbiased environment dynamics learning in confounded POMDPs. Our method is flexible and can be integrated into the model component of off-the-shelf MBRL frameworks without relying on kernel functions or tabular representations.
- **Empirical effectiveness:** Validated on the MuJoCo benchmark and a real-world dataset, our method has shown effective mitigation of confounding bias and superior performance over off-the-shelf standard RL methods. Source code is available at `https://anonymous.4open.science/r/DPC_MBRL-84D5/`

## 2 PRELIMINARY

### 2.1 NOTATION

In this work, we consider an interactive environment modeled as a POMDP represented by the tuple $\mathcal{M} = (\mathcal{S}, \mathcal{A}, \mathcal{R}, \mathcal{U}, P_{\mathcal{R}}, P_{\mathcal{S}}, \rho_0, \gamma, )$, where $\mathcal{S} \subseteq \mathbb{R}^{n_{\mathcal{S}}}$ denotes the observed state space with $S_t \in \mathcal{S}$ being the set of observed states at time $t$ and $s_t$ its realization. Similarly, $\mathcal{U} \subseteq \mathbb{R}^{n_{\mathcal{U}}}$ represents the unobserved confounder space with $U_t \in \mathcal{U}$ being the set of unobserved confounders at time $t$ and $u_t$ the realization, $\mathcal{A} \subseteq \mathbb{R}^{n_{\mathcal{A}}}$ the action space with $A_t \in \mathcal{A}$ being the set of action at time $t$ and $a_t$ the realization, and $\mathcal{R} \subseteq \mathbb{R}$ the set of rewards with $R_t \in \mathcal{R}$ and $r_t$ its realization. A transition step in the environment can be expressed concerning a reward distribution $P_{\mathcal{R}}$ and a dynamics distribution $P_{\mathcal{S}}$ as $R_{t+1} \sim P_{\mathcal{R}}(\cdot|S_t, A_t, U_t)$ and $S_{t+1} \sim P_{\mathcal{S}}(\cdot|S_t, A_t, U_t)$, respectively. Further, initial states are distributed according to $S_0 \sim \rho_0$ and actions according to the policy $A_t \sim \pi(\cdot|S_t, U_t)$. Let $\eta^\pi$ denote the expected cumulative reward under the policy $\pi$ discounted by $\gamma \in [0, 1]$: $V(\pi) = \mathbb{E}_\pi \left[ \sum_{t=0}^{T} \gamma^t R_{t+1} \right]$, referred to as the policy value. We aim to learn an optimal policy $\pi^* = \arg\max_\pi V(\pi)$ that maximizes the policy value.

### 2.2 CHALLENGE IN MODEL LEARNING UNDER CONFOUNDED POMDPS

The first step of MBRL involves learning the dynamics model from observed data (Moerland et al., 2023). In a fully observable setting, this can be done via supervised learning, where the agent learns a forward dynamics model

$$\hat{S}_{t+1} = f(S_t, A_t), \tag{1}$$

to approximate the true transition distribution $P_{\mathcal{S}}$ of the environment. Often, the reward function is also learned, which we omit here for brevity. Once the dynamics and reward functions have been learned, the next step in MBRL is policy learning, where the agent optimizes its decision-making policy based on the learned model. The above procedure is iterated until the cumulative reward converges to the optimal.

While conceptually straightforward, unbiased learning of the dynamics model under a confounded POMDP is inherently difficult since current observations do not provide complete information about the true underlying state. We assume that there exist unobservable confounders $U_t$ in the true environment that may simultaneously affect the action and the next state, and the underlying data-generating process is as follows:

$$A_t \sim \pi(\cdot \mid S_t, U_t), \quad S_{t+1} = h(S_t, A_t, U_t) + \epsilon, \quad \mathbb{E}[\epsilon \mid S_t, A_t, U_t] = 0, \tag{2}$$

where $h$ is an unknown, potentially nonlinear continuous transition function that maps the tuple $(S_t, A_t, U_t)$ to the next state with additive noise $\epsilon$. Under this data-generating process, the observed conditional expectation of $S_{t+1}$ given $A_t = a_t$ and $S_t = s_t$ can be expressed as

$$\mathbb{E}[S_{t+1} \mid S_t = s_t, A_t = a_t] = \int h(s_t, a_t, u_t) \, p(u_t \mid s_t, a_t) \, du_t. \tag{3}$$

However, in the context of RL, what we need is the interventional expectation since taking an action is an intervention, which can be represented as follows using the do-operator, commonly seen in causal inference literature (Pearl, 2009):

$$\mathbb{E}[S_{t+1} \mid S_t = s_t, do(A_t = a_t)] = \int h(s_t, do(a_t), u_t)\, p(u_t \mid s_t)\, du_t. \tag{4}$$

In causal inference, $do(X = x)$ denotes an exogenous intervention on the variable $X$, setting it explicitly to $x$, rather than observing its natural occurrence. Under the intervention $do(A_t = a_t)$, the dependence between $A_t$ and $U_t$ given $S_t$ is severed (i.e., $A_t \perp U_t \mid S_t$), and the distribution changes from $p(u_t \mid s_t, a_t)$ to $p(u_t \mid s_t)$. In general, if $U_t$ is unobserved, $\mathbb{E}[S_{t+1} \mid S_t = s_t, do(A_t = a_t)]$ cannot be identified from $\mathbb{E}[S_{t+1} \mid S_t = s_t, A_t = a_t]$ alone. As a result, $\mathbb{E}[S_{t+1} \mid S_t = s_t, A_t = a_t]$ represents a biased estimation of the true dynamic transition mapping. Based on simulated data generated by such a biased dynamics model learned from the observed states alone, the estimation of the policy value is also biased. Appendix C presents an example of biased trajectories sampled from a biased dynamics model in the Mujoco environment. For notational simplicity, we omit the do-operator in the rest of the paper, with the understanding that the conditional expectation of states or rewards involving actions is to be interpreted causally.

## 3 METHODS

In this section, we introduce the proposed MBRL method for confounded POMDPs. First, we establish an identification result for the policy value by introducing bridge functions with proxy variables, which guarantees consistent identification of the policy value even in confounded POMDPs (Sec. 3.1). Then, we employ deep neural networks to estimate the bridge functions, thereby enhancing flexibility and avoiding restrictions imposed by pre-specified kernel functions or discrete observation spaces (Sec. 3.2). Finally, we present the algorithm of our proposed DPC-MBRL method (Sec. 3.3).

### 3.1 POLICY VALUE IDENTIFICATION

To consistently identify the policy value under confounded POMDPs, we propose to leverage the proximal causal inference framework (Miao et al., 2018; Tchetgen Tchetgen et al., 2024) to eliminate the confounding bias in the model learning process. To adapt proximal causal inference to MBRL, following the framework, we assume that we can additionally observe the so-called outcome-inducing proxy variables $W_t$ that are only related to the action $A_t$ through $(S_t, U_t)$ and action-inducing proxy variables $Z_t$ that are only related to the next state $S_{t+1}$ through $(S_t, U_t)$ at each time $t$, as shown by the directed acyclic graph (DAG) in Figure 2. These proxy variables are required to satisfy the Completeness assumption, which can be interpreted as the proxies being "sufficiently rich":

**Assumption 3.1** (Completeness on Unobserved Confounders (Tchetgen Tchetgen et al., 2024)). *For every $t \in \{1, \ldots, T\}$, $a_t \in \mathcal{A}$, $s_t \in \mathcal{S}$:*

*(i) For any square–integrable function $\phi \in L^2(U_t)$,*

$$\mathbb{E}[\phi(U_t) \mid Z_t,\ A_t = a_t,\ S_t = s_t] = 0 \text{ almost surely} \implies \phi(U_t) = 0 \text{ almost surely.}$$

*(ii) For any square–integrable function $\psi \in L^2(Z_t)$,*

$$\mathbb{E}[\psi(Z_t) \mid W_t,\ A_t = a_t,\ S_t = s_t] = 0 \text{ almost surely} \implies \psi(Z_t) = 0 \text{ almost surely.}$$

Essentially, this assumption requires that the observed proxy variables $\{Z_t, W_t\}$ carry sufficient information about the unobserved confounder $U_t$, thereby shifting the dependence on $U_t$ to $\{Z_t, W_t\}$. This requirement is relatively mild and widely used in statistics and causal inference (Newey & Powell, 2003; Pearl, 2009). Recalling the example of robot-walking in Section 1, we can treat the prior belief about environmental humidity as an action-inducing proxy variable $Z_t$, which captures a prior assessment of how damp the terrain may be before each step and does not directly affect the reward and next state. We can use torque anomalies caused by slipping as an outcome-inducing

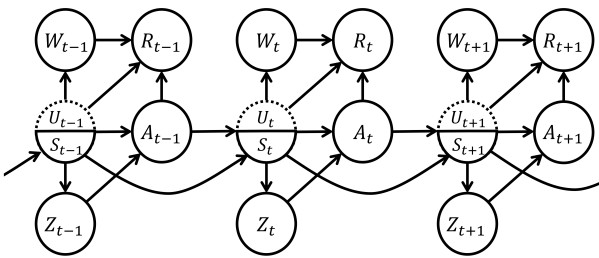

Figure 2: A schematic DAG illustrating the relationships among variables in a confounded POMDP with proxy variables.

proxy $W_t$, which reflects the latent ground slipperiness $U_t$ through $Z_t$ and do not directly affect the action choice. Together, the pair $\{Z_t, W_t\}$ encapsulate information about $U_t$, providing the basis for the identification of the policy value. More details on the selection of proxy variables are provided in Appendix D.1.

Once valid proxy variables are identified, we can leverage proximal causal inference to eliminate the confounding effect of $U_t$. Here, we further assume the existence of certain *bridge functions* under confounded POMDPs in Figure 2:

**Assumption 3.2** (Existence of Reward- and Dynamic-Emission Bridge Functions (Hong et al., 2024; Bennett & Kallus, 2024))**.** *There exist reward-emission bridge functions $h_R$ and dynamic-emission bridge functions $h_D$ that satisfy the following integral equations for each time step $t = 1, \ldots, T$:*

$$p(r_t \mid s_t, a_t, z_t) = \mathbb{E}[h_R(r_t, s_t, a_t, W_t) \mid s_t, a_t, z_t], \quad and,$$
$$p(s_{t+1} \mid s_t, a_t, z_t) = \mathbb{E}[h_D(s_{t+1}, s_t, a_t, W_t) \mid s_t, a_t, z_t].$$

The introduction of reward-emission bridge functions $h_R$ and dynamic-emission bridge functions $h_D$ "bridge" the influence of unobserved confounders $U_t$ into the observable space by leveraging proxy variables $\{Z_t, W_t\}$, thereby enabling estimation of the reward and next state without directly observing $U_t$. Similar versions of this assumption have also been utilized in one recently developed causal inference method called the double negative control (Tchetgen Tchetgen et al., 2024; Miao et al., 2024) and off-policy evaluation methods for confounded POMDPs in the model-free settings (Miao et al., 2022; Shi et al., 2022; Bennett & Kallus, 2024).

Finally, we present the identification results for policy value under confounded POMDPs, as shown in Theorem 3.1 below. A detailed proof is provided in Appendix E.

**Theorem 3.1.** *(Policy Value Identification Results for Confounded POMDPs). Under Assumption 3.1 and 3.2, for each $t = 1, \ldots, T$, the policy value*

$$
\begin{aligned}
V(\pi) &= \sum_{t=1}^{T} \int_{\mathcal{R}} r_t \, p^\pi(r_t) \, dr_t \\
&= \sum_{t=1}^{T} \int r_t \bigg\{ \left( \int h_R(r_t, s_t, a_t, w_t) p(w_t \mid s_t, a_t, z_t) \, dw_t \right) \\
&\quad \times \prod_{j=1}^{t-1} \left( \int h_D(s_{j+1}, s_j, a_j, w_j) p(w_j \mid s_j, a_j, z_j) \, dw_j \right) \\
&\quad \times \left[ \prod_{j=1}^{t} \pi(a_j \mid s_j) p(z_j \mid s_j) \right] \rho(s_1) \bigg\} ds_{1:t} da_{1:t} dz_{1:t} dr_t.
\end{aligned}
\tag{5}
$$

Based on Lemma 1 in Appendix F, the policy value can initially be expressed by decomposing the marginal distribution of the reward $R_t$ at time $t$ under policy $\pi$, i.e. $p^\pi(r_t)$, into the conditional

reward distribution, the environment's transition dynamics, and the policy components. However, the presence of unobserved confounders $U_t$ prevents consistent identification of the policy value. By introducing the reward-emission bridge functions $h_R$ and dynamic-emission bridge functions $h_D$, we can reformulate the expression as a sequential integration of bridge functions with proxy variables, yielding the identifiable representation stated in Theorem 3.1. This sequential integration form enables identification of the policy value without direct access to the unobserved confounders, thereby overcoming confounding in partially observable environments.

*Remark* 1. Our proposed method not only enables the identification of the policy value $V(\pi)$ but also enables next-state transition prediction at each time step. This is an additional advantage that cannot be achieved by model-free RL approaches (Shi et al., 2022) Miao et al. (2024); Bennett & Kallus (2024), since they do not explicitly model or estimate the next state. Such capability is useful in tasks where the objective is not solely to maximize the expected cumulative reward, for example, in risk-sensitive RL (Rigter et al., 2023) and Bayesian RL (Ghavamzadeh et al., 2015), where an accurate characterization of the next-state distribution is essential (Hong et al., 2024).

### 3.2 Estimation of Bridge Functions

In the previous section, we established the consistent identification of the policy value. According to Equation 5, we now need to estimate the bridge functions $h_R$ and $h_D$. Motivated by the work on deep proximal causal inference by Kompa et al. (2022), we employ neural networks to estimate $h_R$ and $h_D$ according to the integral equation in Equation 5. Neural networks are chosen for their flexibility, scalability, and adaptability to diverse data types, which is also a common modeling paradigm for dynamics models in MBRL (Nagabandi et al., 2018; Janner et al., 2019; Yu et al., 2020; Frauenknecht et al., 2024).

First, we reformulate the integral equations in Equation 5 as conditional moment restrictions. For notational convenience, we induce a mean bridge: $\bar{h}_D(s_t, a_t, W_t) := \int s_{t+1} h_D(s_{t+1}, s_t, a_t, W_t) \, ds_{t+1}$. To save space, we only present the formulation for the dynamic-emission bridge function. The reward-emission bridge function can be found in Appendix G. Specifically, we require $\mathbb{E}[S_{t+1} - \bar{h}_D(S_t, A_t, W_t) \mid S_t, A_t, Z_t] = 0$. Then, for any measurable function $g : \mathcal{S} \times \mathcal{A} \times \mathcal{Z} \to \mathbb{R}$, we obtain the unconditional moment condition $\mathbb{E}[(S_{t+1} - \bar{h}_D(S_t, A_t, W_t)) g(S_t, A_t, Z_t)] = 0$. This yields an infinite number of moment restrictions, and following the maximum moment restriction (MMR) framework of Muandet et al. (2020), we define the risk functional by

$$R(\bar{h}_D) = \sup_{\|g\| \leq 1} \left( \mathbb{E}[(S_{t+1} - \bar{h}_D(S_t, A_t, W_t)) g(S_t, A_t, Z_t)] \right)^2, \tag{6}$$

where $\bar{h}_D$ is estimated via a minimax strategy that minimizes the risk $R(\bar{h}_D)$ for the worst-case value of $g$. If $g$ belongs to a reproducing kernel Hilbert space (RKHS), the risk admits the equivalent kernel representation (Mastouri et al., 2021):

$$R_k(\bar{h}_D) = \mathbb{E}\big[(S_{t+1} - \bar{h}_D(S_t, A_t, W_t))(S'_{t+1} - \bar{h}_D(S'_t, A'_t, W'_t)) k\big((S_t, A_t, Z_t), (S'_t, A'_t, Z'_t)\big)\big], \tag{7}$$

where $(S_t, A_t, W_t, Z_t, S_{t+1})$ and $(S'_t, A'_t, W'_t, Z'_t, S'_{t+1})$ are independent copies of the random tuple drawn from $\mathcal{M}$ at time $t$, and $k : (\mathcal{S} \times \mathcal{A} \times \mathcal{Z})^2 \to \mathbb{R}$ is a continuous, bounded, and Integrally Strictly Positive Definite (ISPD) kernel. Then, if $\bar{h}_D$ satisfies $R_k(\bar{h}_D) = 0$, we have $\mathbb{E}[S_{t+1} - \bar{h}_D(S_t, A_t, W_t) \mid S_t, A_t, Z_t] = 0$. Thus, if we can find a neural network parameterization of $\bar{h}_D$ that attains $R_k(\bar{h}_D) = 0$, we will obtain a solution to the dynamic-emission bridge function, which can be used to estimate the next state $S_{t+1}$.

Next, we estimate the neural network parameterization of the dynamic-emission bridge function $\bar{h}_D(\cdot; \theta)$, where $\theta$ denotes the parameters of the neural network. Given samples $\mathcal{D}_n = \{(s^i_t, a^i_t, w^i_t, z^i_t, s^i_{t+1})\}^n_{i=1}$, the empirical risk $\widehat{R}_{k,n}$ can be represented in the form of a V-statistic form over the sample, with an additional $\ell_2$ regularization term:

$$\widehat{R}_{k,n}(\bar{h}_D) = \frac{1}{n^2} \sum_{i=1}^{n} \sum_{j=1}^{n} \varepsilon_i^\top k_{ij} \varepsilon_j, \qquad \widehat{R}_{k,\lambda,n}(\bar{h}_D) = \widehat{R}_{k,n}(\bar{h}_D) + \lambda \|\theta\|_2^2, \tag{8}$$

where $\varepsilon_i$ denotes the prediction residual, defined as $\varepsilon_i = s^i_{t+1} - \bar{h}_D(s^i_t, a^i_t, w^i_t) \in \mathbb{R}^{d_s}$; $k_{ij}$ denotes the evaluation of the kernel on the pair of samples $(s^i_t, a^i_t, z^i_t)$ and $(s^j_t, a^j_t, z^j_t)$, defined as

$k_{ij} = k\big((s_t^i, a_t^i, z_t^i), (s_t^j, a_t^j, z_t^j)\big)$. For implementation, it is convenient to use matrix notation. Let $\Xi = [\varepsilon_1, \ldots, \varepsilon_n]^\top \in \mathbb{R}^{n \times d_s}$ and $K = [k_{ij}] \in \mathbb{R}^{n \times n}$. Then, the neural network parameters $\theta$ can be learned by minimizing the following regularized loss function

$$\mathcal{L}_n(\theta) = \frac{1}{n^2} \operatorname{Tr}\big(\Xi^\top K \Xi\big) + \lambda \|\theta\|_2^2. \tag{9}$$

*Remark* 2. The primary distinction between our work and the existing model-based RL approaches for confounded POMDPs (Hong et al., 2024) lies in the modeling of the bridge functions $h_D$ or $h_R$. In the work of Hong et al. (2024), the bridge functions $h_D$ or $h_R$ is modeled based on a kernel-based identification result, from which a nonparametric two-stage estimation procedure is developed to construct $h_D$ or $h_R$. However, this approach relies on a prespecified kernel function and a two-stage estimation process. Such a design is not data-adaptive and struggles to scale to large datasets, and it also does not align well with the current mainstream practice of modeling dynamics with neural networks in MBRL algorithms (Nagabandi et al., 2018; Kurutach et al., 2018; Janner et al., 2019; Yu et al., 2020; Frauenknecht et al., 2024). In contrast, our method employs neural network–based bridge function estimation, providing a data-adaptive and scalable solution that aligns with mainstream neural network approaches to modeling transition dynamics and avoids restrictions to dependence on prespecified kernels or multi-stage estimation.

### 3.3 Model-based policy optimization

Once we have obtained the trained dynamics-emission bridge function $\hat{h}_D$ and reward-emission bridge functions $\hat{h}_R$, we can employ them to rollout simulated trajectories starting from the current state $s_t$ and the agent-selected action $a_t$. By recursively applying $\hat{h}_D$ and $\hat{h}_R$ to generate the next states and reward, we can construct model-based rollouts that serve as simulated experiences for policy optimization and planning.

To demonstrate that our method can be used for policy optimization and planning, we integrate it into a standard Dyna-style MBRL framework (Liu & Wang, 2021; Dong et al., 2024), as shown in Algorithm 1. Our method modifies and enhances Lines 4 and 6. In Line 3, the agent interacts with the real environment under policy $\pi$ to collect data. However, because the environment contains unobserved confounders, the dynamics model trained directly on these data in Line 4 becomes biased, which further leads to biased simulated trajectories $\tilde{\tau}$ in Line 6.

To address this issue, we introduce proxy variables and bridge functions to recover unbiased dynamics and reward models using confounded data (Line 4). With these recovered models, the simulated rollouts generated in Line 6 provide consistent estimates of both the next state and reward at every time step, rather than confounded predictions. Theorem 3.1 guarantees that, under policy $\pi$, rolling out the learned model yields consistent transition and reward estimates. Since Lines 6–8 operate entirely on deconfounded models, they do not reintroduce confounding, and no further deconfounding is required.

---

**Algorithm 1** Deep Proximal Causal MBRL for Confounded POMDPs

---

1: Initialize policy $\pi$, bridge functions $h_D$ and $h_R$; empty buffers $\mathcal{D}_{\text{env}}$, $\mathcal{D}_{\text{mod}}$; real environment $\mathcal{M}$.
2: **repeat**
3:     Collect real trajectories $\tau$ from the real environment $\mathcal{M}$ under $\pi$ and add $\tau$ to buffer $\mathcal{D}_{\text{env}}$.
4:     Estimate bridge functions $h_D$ and $h_R$ on $\mathcal{D}_{\text{env}}$ by Eq. 9 and Eq. 24.
5:     **repeat**
6:         Generate simulated trajectories $\tilde{\tau}$ from $\hat{h}_D$ and $\hat{h}_R$ under $\pi$ by Eq.5 and add $\tilde{\tau}$ to buffer $\mathcal{D}_{\text{mod}}$.
7:         Update policy $\pi$ on $\mathcal{D}_{\text{mod}}$ (e.g., policy gradient / actor–critic).
8:         Estimate performance $V(\pi)$.
9:     **until** stopping criterion is met.
10: **until** the policy $\pi$ performs well in the real environment $\mathcal{M}$.
11: **return** policy $\pi$

---

## 4 EXPERIMENTS

In this section, we conduct two experiments: an advanced physics simulation MuJoCo and a real-world dataset to demonstrate the effectiveness of DPC-MBRL.

### 4.1 SIMULATED EXPERIMENT

#### 4.1.1 EXPERIMENTAL SETUP

First, we evaluate our proposed approach on the D4RL benchmark (Fu et al., 2021), a suite of Gym-MuJoCo environments comprising several continuous control tasks. This benchmark offers a variety of synthetic robotic agents ranging from simple locomotion to complex manipulation, and is characterized by high-dimensional state spaces and long-horizon control challenges.

In this experiment, we select four environments: Swimmer, Hopper, HalfCheetah, and Ant, to evaluate our method on cases with a diverse range of agent state dimensions, from a few to around a hundred. The details of each environment be found in Appendix H.1. To create partially observable environments, we follow previous studies (Miao et al., 2022; Shi et al., 2022) and simulate unobserved confounders by adding Gaussian noises that simultaneously affect the agent's actions and the next states at each time step. During model training, these confounders are hidden. The details of data generation are presented in Appendix H.2.

Our method (DPC-MBRL) is plugged into several state-of-the-art MBRL algorithms (which do not have deconfounding abilities), MBPO (Janner et al., 2019), M2AC (Pan et al., 2020), and MACURA (Frauenknecht et al., 2024), to demonstrate that our algorithm enables the agent to learn the true transition dynamics unbiasedly, even in the presence of unobserved confounders in the environment.

#### 4.1.2 RESULTS

Table 1 (and Table 3 in Appendix H.3 ) presents the model's estimation error across different simulated trajectory lengths as the number of environment interaction steps gradually increases. The estimation error is defined as the MSE at each rollout step between the state sequences generated by the learned model and the true environment under the same policy. Overall, the three standard MBRL algorithms augmented with our deconfounding approach consistently achieve lower MSEs than their original or confounded version, which do not account for unobserved confounders. This demonstrates that our method enables MBRL algorithms to learn environment dynamics with greater fidelity. To save space, Table 1 below only shows the results of MBPO and the full results are provided in Appendix H.3.

Table 1: Estimation error (mean $\pm$ std) across MuJoCo benchmark (full table in Appendix H.3).

| Methods | Time steps (Rollout lengths) | Mojuco tasks | | | |
|---------|------------------------------|---------|--------|---------------|-----|
| | | Swimmer | Hopper | Half Cheetah | Ant |
| Confounded MBPO | 1 k ( 1-step) | $0.85 \pm 0.78$ | $1.24 \pm 1.40$ | $3.31 \pm 2.65$ | $8.69 \pm 4.46$ |
| | 10k ( 5-step) | $0.95 \pm 0.81$ | $1.47 \pm 1.48$ | $8.10 \pm 4.52$ | $9.36 \pm 5.04$ |
| | 20k (10-step) | $0.95 \pm 0.84$ | $1.65 \pm 1.41$ | $12.63 \pm 7.28$ | $12.72 \pm 6.09$ |
| | 30k (15-step) | $1.32 \pm 1.08$ | $2.14 \pm 1.79$ | $21.01 \pm 12.19$ | $24.30 \pm 14.15$ |
| DPC-MBPO | 1 k ( 1-step) | $0.18 \pm 0.13$ | $0.13 \pm 0.15$ | $1.48 \pm 0.81$ | $5.11 \pm 1.82$ |
| | 10k ( 5-step) | $0.23 \pm 0.13$ | $0.32 \pm 0.26$ | $6.17 \pm 3.25$ | $6.44 \pm 2.57$ |
| | 20k (10-step) | $0.31 \pm 0.16$ | $0.63 \pm 0.43$ | $8.07 \pm 4.28$ | $7.06 \pm 2.97$ |
| | 30k (15-step) | $0.54 \pm 0.39$ | $1.58 \pm 1.20$ | $8.63 \pm 4.64$ | $9.13 \pm 3.78$ |

## 4.2 REAL WORLD EXPERIMENT

### 4.2.1 EXPERIMENTAL SETUP

Next, we consider a more "real-world" inspired application. Specifically, we demonstrate the effectiveness of the proposed method using a real dataset originally designed to evaluate the impact of right heart catheterization (RHC) on critically ill patients in intensive care units (ICUs), obtained from the Study to Understand Prognoses and Preferences for Outcomes and Risks of Treatments (SUPPORT) (Ben-Moshe & Curseen, 2023). This dataset has been extensively analyzed in the literature on causal inference and offline policy evaluation (OPE). A more detailed description of this dataset, including unobserved confounders and the selection of relevant proxy variables can be found in Appendix I.

In the experiment, we consider four approaches. The first is a vanilla MBRL method that ignores the presence of unobserved confounders and directly trains the model using all observed states as inputs. The second and third are two state-of-the-art model-free algorithms for confounded POMDPs, denoted as pFQE (Miao et al., 2022) and PCI (Bennett & Kallus, 2024), both of which leverage techniques from proximal causal inference to address unobserved confounders in the MDP. The fourth is our proposed method.

### 4.2.2 RESULTS

In this experiment, the objective is to estimate the expected survival time of patients under the treatment RHC. For each setting of proxy variables, we independently conduct 20 runs of every method and visualize the distribution of the outcomes across repeated trials with boxplots. Figure 3 presents the comparative results between our method and the aforementioned approaches.

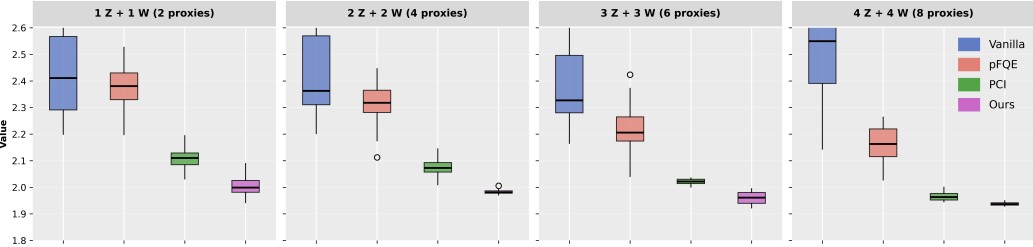

Figure 3: Boxplots of logMSE with respect to the ground truth for different methods (Vanilla, pFQE, PCI, and ours) under varying numbers of proxy variables (2, 4, 6, and 8).

As shown in Figure 3, our method outperforms the alternatives across all the proxy-variable settings. In contrast, the vanilla approach, which ignores the presence of confounders, exhibits substantially larger variance and inaccurate estimates. The model-free RL methods that leverage proxies, pFQE, and PCI, yield relatively more accurate results, but still fall short of the accuracy and low variance attained by our proposed method. By comparison, our method consistently achieves the lowest error and the most stable performance across all scenarios, enabling a reliable identification of policy values even in the presence of unobserved confounders.

Moreover, focusing on the same class of proxy-based approaches (i.e., pFQE, PCI, and ours), we observe that the performance improves as the number of proxies increases. This trend supports the Completeness Assumption 3.1: the more proxies are available, the more information they carry about the unobserved confounders, enhancing the representation and recovery of hidden variables, and consequently leading to more accurate and stable estimation.

## 5 CONCLUSION

To overcome the bias caused by unobserved confounders in the environment, we propose an MBRL method for confounded POMDPs. Under some mild assumptions, we establish a consistent identification result for policy value by using bridge functions with proxy variables. The identification

result guarantees an unbiased estimation of the policy values even in the presence of unobserved confounders in the environment. We empirically demonstrate the effectiveness of our method using an advanced physics simulation environment and a real-world medical dataset.

Although our method shows promise, it still has several limitations. For example, in real-world scenarios, identifying valid proxies from the available observable variables may not be straightforward, often requiring substantial prior knowledge. This also opens up several meaningful directions for future work, such as the development of knowledge engineering or representation learning techniques to extract valid proxies from the observed variables effectively.

ETHICS STATEMENT

Our study uses only publicly available benchmark datasets (MuJoCo control tasks (Fu et al., 2021) and the RHC ICU dataset [1]) and does not involve identifiable human subjects or sensitive personal data. All experimental protocols follow the ICLR Code of Ethics[2]. No potential conflicts of interest or security/privacy concerns have been identified.

REPRODUCIBILITY STATEMENT

All code, configuration files, and instructions required to reproduce the experiments are provided in an anonymous repository at `https://anonymous.4open.science/r/DPC_ MBRL-84D5/`. The repository includes: (i) scripts for data preprocessing and environment setup, (ii) training and evaluation code for all algorithms and baselines, (iii) configuration files specifying hyperparameters and random seeds, and (iv) documentation describing how to run the experiments and generate the figures and tables reported in the paper. These materials enable independent researchers to fully replicate our results.

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

## A    USE OF LARGE LANGUAGE MODELS (LLMs)

We used an LLM (OpenAI ChatGPT, GPT-5) only as a general-purpose writing and editing assistant. Specifically, it helped with

- polishing grammar and improving clarity of English exposition,
- generating LaTeX equations from given mathematical expressions and checking LaTeX formatting issues,
- generating part of the content in Figure 1,
- drafting this section, ethics statement and reproducibility statement.

The LLM did not contribute to research ideation, algorithm design, data analysis, or experimental results. All conceptual contributions, experiments, and final decisions were made by the authors, who take full responsibility for the content of this paper.

## B    RELATED WORK

To address the confounding effects introduced by unobserved confounders in MDP, one of the most widely adopted strategies is to draw upon causal inference theory for the identification and estimation of policy values (Deng et al., 2023; Zeng et al., 2024). Zhang & Bareinboim (2016) was the first to demonstrate that standard RL algorithms cannot be guaranteed to learn an optimal policy when unobserved confounders are present in the MDP, and it articulated this issue within the language of causality. Moreover, it highlighted causal inference as an effective framework for resolving such challenges. Subsequently, Tennenholtz et al. (2020) referred to such MDPs with unobserved confounding states as confounded POMDPs. Building on the ideas of Miao et al. (2018) in the field of causal inference, Tennenholtz et al. (2020) employed proxy variables to enable the estimation of policy values under the target policy. Since then, a growing body of research has leveraged causal inference techniques to address policy evaluation in various settings of confounded POMDPs. Broadly speaking, these approaches can be categorized into two classes: model-free methods and model-based methods.

We start by providing a review of model-free methods. Wang et al. (2021) leveraged the idea of the backdoor criterion (Pearl, 2009) to iteratively deconfound the policy value function. This constitutes one of the earlier attempts to incorporate the idea of deconfounding into the iterative process of Bellman-type equations. Subsequently, Miao et al. (2022) addressed the challenge of policy evaluation in confounded POMDPs with continuous state spaces by establishing a nonparametric identification result. Specifically, it proposed to iteratively estimate policy values by combining the fitted Q evaluation function (Le et al., 2019) with a bridge function (Tchetgen Tchetgen et al., 2024). In the same year, Shi et al. (2022) also investigated policy evaluation in confounded POMDPs with continuous observation and state spaces, and likewise derived a nonparametric identification result. Unlike Miao et al. (2022), however, Shi et al. (2022) integrated a doubly robust estimator, which combines a value-function-based estimator (Munos & Szepesvári, 2008) and an importance sampling estimator (Liu et al., 2018), with bridge functions (Tchetgen Tchetgen et al., 2024) to estimate the target policy in confounded POMDPs. The researchers in the same group then advanced this line of work by introducing a novel approach that employs a mediator variable (Pearl, 2009) from causal inference to address confounding in POMDPs (Shi et al., 2024). Bennett & Kallus (2024) further extended the proxy-variable-based deconfounding idea Miao et al. (2018) by proposing the proximal reinforcement learning framework, which aims to address unobserved confounding in POMDPs. All the aforementioned methods share a common characteristic: they are model-free approaches, which estimate the policy value directly by solving a series of Bellman-type equations, such as the value function or the Q-function. However, they overlook the extraction of information from the reward function and the dynamics model, leaving a gap in model-based RL algorithms for confounded POMDPs.

A key aspect of model-based RL algorithms for confounded POMDPs lies in establishing identification results of the dynamics transitions and reward function in the confounded POMDP. To the best of our knowledge, two recent studies have made attempts in this area. Gasse et al. (2023) imported ideas from the well-established framework of do-calculus (Pearl, 2009) to express model-based RL

as a causal inference problem and proposed a generic method for learning a causal transition model. However, their work is based on a discrete observation space, which requires learning tabular dynamics models and thus cannot apply to settings with continuous observation/state spaces. Hong et al. (2024), inspired by Singh et al. (2019) Mastouri et al. (2021), developed a two-stage nonparametric estimation procedure for estimating dynamics transitions. Such nonparametric estimation procedure will allow general function approximations, thereby relaxing the restriction of the tabular setting in Gasse et al. (2023). However, this work critically relies on prespecified kernel functions, which undermines data adaptivity and restricts scalability to large datasets. Moreover, in conventional MBRL algorithms (Nagabandi et al., 2018; Kurutach et al., 2018; Janner et al., 2019; Yu et al., 2020; Frauenknecht et al., 2024), the dynamics are typically modeled with neural networks rather than kernel functions, which further hinders the scalability of this method.

## C  A Confounded Environment of MuJoCo

The environment dynamics learned from confounded POMDPs are inherently biased if the unobserved confounders are ignored, leading to biased simulated trajectories when these dynamics are used for sampling. Figure 4 shows the distributional shift among the trajectories sampled from the biased dynamics model, unbiased dynamics model, and the true environment under the same policy. The x-axis denotes the time step $t$. The y-axis denotes the temporal transition of one of the MuJoCo Walker's state variables. The shaded area denotes the 95% confidence interval estimated from 50 independent trajectories.

Because of the influence of unobserved confounders in the environment, the trajectories generated from the biased dynamics model (red) display severe deviation from the ground truth distribution of environment rollouts (green). In contrast, when the same dynamics model is trained from the same environment without unobserved confounders, the simulated trajectories (orange) closely align with the true distribution.

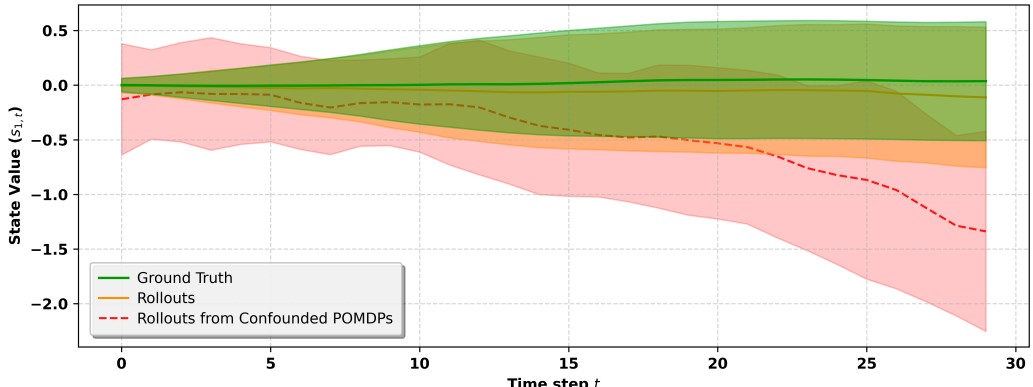

Figure 4: The deviation in the trajectories sampled from the learned dynamics model can be attributed to unobserved confounders in the environment.

## D  Additional Notes on Proxy Variables

### D.1  Different choices of proxy variables

Here, we provide several options on how to choose proxy variables $W_t$ and $Z_t$ that satisfy Assumptions 3.1. Existing representative studies (Miao et al., 2022; Bennett & Kallus, 2024) have systematically discussed the selection of proxy variables. Building on these works, we further distill their core ideas and, from a practical perspective, propose a feasible selection scheme.

**Choice of $W_t$.** We identify two practical categories to select the proxy variable $W_t$. The first is reward/next-state–inducing proxies, which are environmental variables correlated with the outcome

$R_t/S_{t+1}$ but unaffected by the action $A_t$. The second is non-inducing proxies, which only have a causal relationship with the unobserved confounders $U_t$ but have no direct effect on the outcome $R_t/S_{t+1}$. Examples include external disturbances that do not alter the current reward / next state but convey information about $U_t$. Both categories satisfy Assumptions 3.1. Figure 5 illustrates several possible causal relationships among $W_t$, $S_t$, $U_t$, and $R_t$, where a causal relationship between $U_t$ and $W_t$ is required, while the effect of $W_t$ on $R_t$ is optional.

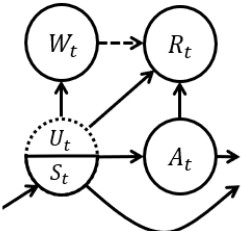

Figure 5: Causal relationship about $W_t$. Dashed arrows: optional causal effect. $W_t$ may or may not affect $R_t$.

**Choice of $Z_t$.** Once $W_t$ is determined, there are several valid choices of $Z_t$ that are compatible with $W_t$. One option is to identify, at the current time step, a variable from the environmental variables that is associated with the action $A_t$ but does not directly influence $W_t$ and $S_{t+1}/R_t$ (See Figure 6). The second option, which is more intuitive, is to define $Z_t$ based on the observed history (See Figure 7). Within this framework, three practical choices can be considered: 1) **Previous-step observation.** We use previous-step observation as proxies for $U_t$, i.e., $Z_t = (S_{t-1}, W_{t-1}, A_{t-1}, R_{t-1})$. It is easy to verify that this satisfies Assumption 3.1. This kind of choice may be preferable in applications where the environment is approximately Markovian and short-term dependencies dominate, so that the most recent observation captures nearly all the information needed for estimating the next state and reward, for example, continuous-control tasks or real-time process control. 2) **Full-history aggregation.** We use the entire historical observation as the proxies for $U_t$, i.e., $Z_t = (S_0, W_0, A_0, R_0, \ldots, S_{t-1}, W_{t-1}, A_{t-1}, R_{t-1})$. This kind of choice may be preferable in applications where the system exhibits long-range temporal dependencies or strong non-Markovian dynamics, making the entire past trajectory essential for accurate identification, for example, longitudinal financial analysis. In such settings, the task essentially reduces to a time-series prediction problem. This choice, however, faces two drawbacks: (i) substantially increased computational burden, and (ii) accumulation of multi-step estimation errors. 3) $k-$**prior observation.** We use the $k-$most recent time steps observation as proxies for $U_t$, i.e., $Z_t = (S_{t-k:t-1}, W_{t-k:t-1}, A_{t-k:t-1}, R_{t-k:t-1})$. This approach can be viewed as a natural trade-off between the single-step history and the full-history aggregation.

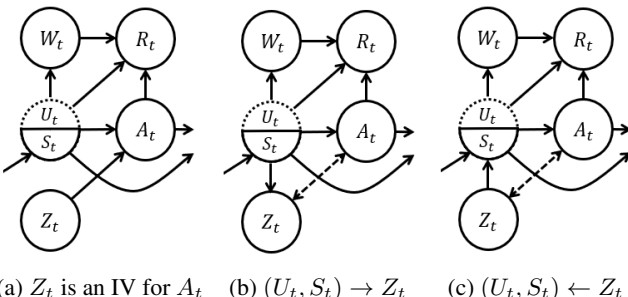

(a) $Z_t$ is an IV for $A_t$    (b) $(U_t, S_t) \to Z_t$    (c) $(U_t, S_t) \leftarrow Z_t$

Figure 6: Causal relationship about $Z_t$. In panel (a), $Z_t$ is unaffected by unobserved confounders $U_t$ and essentially functions as an instrumental variable Newey & Powell (2003). In panels (b) and(c), the dashed arrows denote an optional causal relationship, $Z_t \to A_t$, $A_t \to Z_t$, or no causal effect.

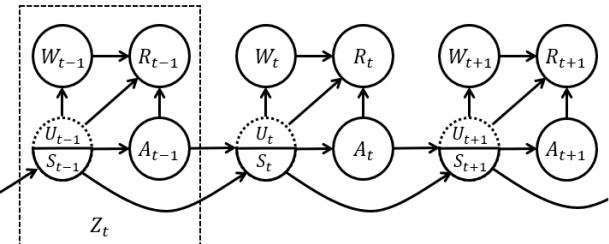

Figure 7: An example of $Z_t$ as the previous observation.

# E    PROOF OF THEOREM 3.1

In this section, we present a complete proof of the identification results summarized in Theorem 3.1.

By the definition of the policy value,

$$\mathcal{V}(\pi) = \mathbb{E}^\pi \left[ \sum_{t=1}^T R_t \right] = \sum_{t=1}^T \int_\mathcal{R} r_t \, p^\pi(r_t) \, dr_t. \tag{10}$$

It remains to identify the marginal distribution of the reward $R_t$ induced by the policy $\pi$, that is, $p^\pi(r_t)$. We express $p^\pi(r_t)$ in terms of the combination of policy functions $\pi_t(a_t \mid s_t)$, the reward-emission models $p(r_t \mid s_t, a_t)$, and the dynamics-emission models $p(s_{t+1} \mid s_t, a_t)$ by unrolling the law of total probability along the trajectory:

$$p^\pi(r_t) = \int_{\mathcal{S}^t} \int_{\mathcal{A}^t} \rho(s_1) \left[ \prod_{k=1}^{t-1} \pi_k(a_k \mid s_k) \, p(s_{k+1} \mid s_k, a_k) \right] \pi_t(a_t \mid s_t) \, p(r_t \mid s_t, a_t) \, da_{1:t} \, ds_{1:t}. \tag{11}$$

This follows from Lemma 1. The above derivation holds for a fully observed, unconfounded MDP. In confounded POMDPs, we rewrite the kernels to depend on $U_t$. Let the initial distribution be $\mu(s_1, u_1)$, the policy functions $\pi_t(a_t \mid s_t, u_t)$, the dynamics-emission models $p(s_{t+1}, u_{t+1} \mid s_t, a_t, u_t)$, and the reward-emission models $p(r_t \mid s_t, a_t, u_t)$. Then, under policy $\pi$, we have:

$$p^\pi(r_t) = \int_{\mathcal{S}^t} \int_{\mathcal{A}^t} \int_{\mathcal{U}^t} \mu(s_1, u_1) \left[ \prod_{k=1}^{t-1} \pi_k(a_k \mid s_k, u_k) \, p(s_{k+1}, u_{k+1} \mid s_k, a_k, u_k) \right]$$
$$\times \pi_t(a_t \mid s_t, u_t) \, p(r_t \mid s_t, a_t, u_t) \, da_{1:t} \, ds_{1:t} \, du_{1:t}. \tag{12}$$

The above formulation does not permit unbiased estimation, as the influence of the unobserved confounders $U_t$ on both the reward-emission models and the dynamics-emission models cannot be integrated out.

To address this issue, we introduce proxy variables to mitigate the bias induced by unobserved confounders. Based on DAG 2 with proxies , we can rewrite Eq. 12 as:

$$p^\pi(r_t) = \int_{\mathcal{S}^t} \int_{\mathcal{A}^t} \int_{\mathcal{U}^t} \mu(s_1, u_1) \prod_{k=1}^{t-1} \left\{ \int_\mathcal{Z} p(z_k \mid s_k, u_k) \, \pi_k(a_k \mid s_k, u_k) \right.$$
$$\left. \times \int_\mathcal{W} p(w_k \mid s_k, a_k, z_k, u_k) \, p(s_{k+1}, u_{k+1} \mid s_k, a_k, w_k, u_k) \, dw_k \, dz_k \right\}$$
$$\times \int_\mathcal{Z} p(z_t \mid s_t, u_t) \, \pi_t(a_t \mid s_t, u_t) \int_\mathcal{W} p(w_t \mid s_t, a_t, z_t, u_t)$$
$$\times p(r_t \mid s_t, a_t, w_t, u_t) \, dw_t \, dz_t \, da_{1:t} \, ds_{1:t} \, du_{1:t}. \tag{13}$$

To obtain an unbiased estimate of this reward distribution, we decompose the task into two components: (i) estimation of the immediate reward at the current time step; and (ii) estimation of the state-transition dynamics from the initial state through the preceding time step.

**Step 1.** First, we estimate the immediate reward at the current time step by using the reward-emission bridge function. Compute the innermost integral over $w_t$ and drop $z_t$ based on $R_t \perp Z_t \mid (S_t, A_t, U_t)$:

$$\int p(w_t \mid s_t, a_t, z_t, u_t)\, p(r_t \mid s_t, a_t, w_t, u_t)\, dw_t = p(r_t \mid s_t, a_t, z_t, u_t) = p(r_t \mid s_t, a_t, u_t). \quad (14)$$

By introducing the reward-emission bridge functions $h_R$ in Assumption 3.2, the conditional distribution of reward can be written as

$$p(r_t \mid s_t, a_t, u_t) = \mathbb{E}\big[\mathbb{E}[h_R(r_t, s_t, a_t, W_t) \mid s_t, a_t, z_t] \mid s_t, a_t, u_t\big] = \mathbb{E}[h_R(r_t, s_t, a_t, W_t) \mid s_t, a_t, u_t].$$

Substituting this expression back into Eq. 13, we obtain

$$\mathcal{I}_t = \int_{\mathcal{S}} \int_{\mathcal{Z}} \int_{\mathcal{A}} \int_{\mathcal{U}} p(z_t \mid s_t, u_t)\, \pi_t(a_t \mid s_t, u_t)\, \underbrace{\mathbb{E}[\,h_R(r_t, s_t, a_t, W_t) \mid s_t, a_t, u_t\,]}_{=:\, g_{s_t, a_t}(u_t)}\, p(u_t \mid s_t)$$

$$\times p(s_t)\, du_t\, da_t\, dz_t\, ds_t. \quad (15)$$

where $g_{s_t, a_t}(u_t)$ is a shorthand for the reward-emission bridge function, viewed as a function of $u_t$ with $(s_t, a_t)$ fixed. To save space, only the reward block $\mathcal{I}_t$ is retained, and all unrelated outer layers are omitted. Here we consider a more general setting in which, conditional on $(U_t, S_t)$, $Z_t$ and $A_t$ are independent, as illustrated in Figure 6b. Under this setting, we combine the policy distribution and the action-inducing proxy distribution into the joint distribution $p(z_t, a_t \mid s_t, u_t)$ for notational simplicity, i.e., $p(z_t \mid s_t, u_t)\, \pi_t(a_t \mid s_t, u_t) = p(z_t, a_t \mid s_t, u_t)$. And applying the Bayes rule, we obtain:

$$\mathcal{I}_t = \int \int \int \int p(z_t, a_t \mid s_t, u_t)\, g(u_t)\, p(u_t \mid s_t)\, p(s_t)\, du_t\, da_t\, dz_t\, ds_t$$

$$= \int \int \int \int g(u_t)\, p(z_t, u_t, a_t \mid s_t)\, p(s_t)\, du_t\, da_t\, dz_t\, ds_t$$

$$= \int \int \int \int g(u_t)\, p(u_t \mid s_t, z_t, a_t)\, p(z_t, a_t \mid s_t)\, p(s_t)\, du_t\, da_t\, dz_t\, ds_t$$

$$= \int \int \int p(z_t, a_t \mid s_t)\, p(s_t) \int g(u_t)\, p(u_t \mid s_t, z_t, a_t)\, du_t\, da_t\, dz_t\, ds_t$$

$$= \int \int \int p(z_t, a_t \mid s_t)\, p(s_t)\, \mathbb{E}\big[g(U_t) \mid s_t, z_t, a_t\big]\, da_t\, dz_t\, ds_t. \quad (16)$$

Substituting $g_{s_t, a_t}(u_t) = \mathbb{E}[h_R(r_t, s_t, a_t, W_t) \mid s_t, a_t, u_t]$ into the preceding expression and applying the independence condition $(W_t, R_t) \perp Z_t \mid (S_t, A_t, U_t)$ and Law of Iterated Expectations (LIE), we obtain:

$$\mathbb{E}\big[g(U_t) \mid s_t, z_t, a_t\big] = \mathbb{E}\big[\,\mathbb{E}[h_R(r_t, s_t, a_t, W_t) \mid s_t, a_t, U_t] \mid s_t, z_t, a_t\big]$$

$$= \mathbb{E}\big[\,\mathbb{E}[h_R(r_t, s_t, a_t, W_t) \mid s_t, z_t, a_t, U_t] \mid s_t, z_t, a_t\big]$$

$$= \mathbb{E}\big[h_R(r_t, s_t, a_t, W_t) \mid s_t, a_t, z_t\big]. \quad (17)$$

Then, we have

$$\mathcal{I}_t = \int \int \int p(z_t, a_t \mid s_t)\, p(s_t)\, \mathbb{E}\big[h_R(r_t, s_t, a_t, W_t) \mid s_t, a_t, z_t\big]\, da_t\, dz_t\, ds_t. \quad (18)$$

**Step 2.** We apply the same reasoning to estimate the state-transition dynamics from the initial state to the previous time step. The key principle is to introduce dynamic-emission bridge functions that iteratively removes the influence of $U_t$. Substituting the dynamic-emission bridge function into

Eq. 13, we obtain:

$$
\mathcal{I}_k = \iiiint p(z_k \mid s_k, u_k)\, \pi_k(a_k \mid s_k, u_k)\, \mathbb{E}\big[h_D(s_{k+1}, s_k, a_k, W_k) \mid s_k, a_k, u_k\big]\, p(u_k \mid s_k)
$$
$$
\times\, p(s_k)\, \mathrm{d}u_k\, \mathrm{d}a_k\, \mathrm{d}z_k\, \mathrm{d}s_k.
$$
$$
= \iiint p(z_k, a_k \mid s_k)\, p(s_k) \mathbb{E}\big[h_D(s_{k+1}, s_k, a_k, W_k) \mid s_k, a_k, z_k\big]\, \mathrm{d}a_k\, \mathrm{d}z_k\, \mathrm{d}s_k.
$$

$$(19)$$

**Step 3.** For all $k = 1, \ldots, t-1$, successively substitute Eq. 19 together with Eq. 18 into Eq. 13. We can obtain:

$$
p^\pi(r_t) = \iiint \rho(s_1)\left[\prod_{k=1}^{t-1} p(z_k, a_k \mid s_k)\, \mathbb{E}\big[h_D(s_{k+1}, s_k, a_k, W_k) \mid s_k, a_k, z_k\big]\right]
$$
$$
\times p(z_t, a_t \mid s_t)\, \mathbb{E}\big[h_R(r_t, s_t, a_t, W_t) \mid s_t, a_t, z_t\big]\, dz_{1:t}\, da_{1:t}\, ds_{1:t}. \qquad (20)
$$

Substituting this expression into Eq.10 and rearranging the order of integration yields:

For each $t \in \{1, \ldots, T\}$,

$$
V(\pi) = \sum_{t=1}^{T} \int_{\mathcal{R}} r_t\, p^\pi(r_t)\, dr_t
$$
$$
= \sum_{t=1}^{T} \int r_t \Bigg\{ \left( \int h_R(r_t, s_t, a_t, w_t) p(w_t \mid s_t, a_t, z_t)\, dw_t \right)
$$
$$
\times \prod_{j=1}^{t-1} \left( \int h_D(s_{j+1}, s_j, a_j, w_j) p(w_j \mid s_j, a_j, z_j)\, dw_j \right)
$$
$$
\times \left[ \prod_{j=1}^{t} \pi(a_j \mid s_j) p(z_j \mid s_j) \right] \rho(s_1) \Bigg\} ds_{1:t} da_{1:t} dz_{1:t} dr_t. \qquad (21)
$$

## F   LEMMAS

*Lemma* 1 (Policy value under a MDP). Under the standard Markov decision process setting, where the next state and reward depend only on the current state–action pair, the value of a policy $\pi$ over a finite horizon $T$ is

$$
V(\pi) = \sum_{t=1}^{T} \int_{\mathbb{R}} r_t\, p^\pi(r_t)\, dr_t, \qquad (22)
$$

where $p^\pi(r_t)$ denotes the marginal density of the time-$t$ reward. Moreover, $p^\pi(r_t)$ admits the integral representation

$$
p^\pi(r_t) = \int_{S^t} \int_{A^t} \rho(s_1) \left[ \prod_{k=1}^{t-1} \pi_k(a_k \mid s_k)\, p(s_{k+1} \mid s_k, a_k) \right] \pi_t(a_t \mid s_t)\, p(r_t \mid s_t, a_t)\, da_{1:t}\, ds_{1:t}, \qquad (23)
$$

where $(s_{1:t}, a_{1:t}, r_t)$ be a length-$t$ trajectory generated by policy $\pi$ with initial state distribution $\rho$, transition kernel $p(s_{k+1} \mid s_k, a_k)$, and reward density $p(r_t \mid s_t, a_t)$. For simplicity, we omit the discount factor.

We provide the corresponding proof.

*Proof.* Starting from the joint density of $(r_t, s_t, a_t)$ under policy $\pi$,

$$p^\pi(r_t) = \iint p^\pi(r_t, s_t, a_t)\, da_t\, ds_t$$

$$= \iint p(r_t \mid s_t, a_t)\, p^\pi(s_t, a_t)\, da_t\, ds_t$$

$$= \iint p(r_t \mid s_t, a_t)\, \pi_t(a_t \mid s_t)\, p^\pi(s_t)\, da_t\, ds_t.$$

By repeatedly applying the law of total probability and the Markov property,

$$p^\pi(s_t) = \iint p(s_t \mid s_{t-1}, a_{t-1})\, \pi_{t-1}(a_{t-1} \mid s_{t-1})\, p^\pi(s_{t-1})\, da_{t-1}\, ds_{t-1}$$

$$\vdots$$

Unfolding this recursion down to the initial state yields

$$p^\pi(r_t) = \int_{S^t}\!\int_{A^t} \rho(s_1) \left[\prod_{k=1}^{t-1} \pi_k(a_k \mid s_k)\, p(s_{k+1} \mid s_k, a_k)\right] \pi_t(a_t \mid s_t)\, p(r_t \mid s_t, a_t)\, da_{1:t}\, ds_{1:t},$$

which completes the proof. $\qquad\square$

## G   REWARD-EMISSION BRIDGE FUNCTION

Following the same reasoning as for the dynamic-emission bridge function, we impose the conditional moment restriction

$$\mathbb{E}\big[R_t - h_R(S_t, A_t, W_t) \,\big|\, S_t, A_t, Z_t\big] = 0.$$

Consequently, for every measurable test function $g : \mathcal{S} \times \mathcal{A} \times \mathcal{Z} \to \mathbb{R}$,

$$\mathbb{E}[(R_t - h_R(S_t, A_t, W_t))\, g(S_t, A_t, Z_t)] = 0,$$

which yields an infinite family of unconditional moment equations.

Adopting the maximum–moment matching framework, we define the risk functional

$$R(h_R) = \sup_{\|g\| \le 1} \big(\mathbb{E}[(R_t - h_R(S_t, A_t, W_t))\, g(S_t, A_t, Z_t)]\big)^2.$$

If $g$ lies in a reproducing kernel Hilbert space with a continuous integrally strictly positive definite kernel $k$, this risk admits the equivalent kernel representation

$$R_k(h_R) = \mathbb{E}\Big[(R_t - h_R(S_t, A_t, W_t))(R'_t - h_R(S'_t, A'_t, W'_t))k\big((S_t, A_t, Z_t), (S'_t, A'_t, Z'_t)\big)\Big].$$

Thus $R_k(h_R) = 0$ implies $\mathbb{E}[R_t - h_R(S_t, A_t, W_t) \mid S_t, A_t, Z_t] = 0$ almost surely with respect to $P_{S_t, A_t, Z_t}$. Given samples $\{(s_t^i, a_t^i, w_t^i, z_t^i, r_t^i)\}_{i=1}^n$, the empirical regularized loss used to fit a neural parameterization $h_R(\cdot; \theta)$ is

$$\mathcal{L}_n^{(R)}(\theta) = \frac{1}{n^2}\, \mathrm{Tr}\big(\Xi_R^\top K \Xi_R\big) + \lambda \|\theta\|_2^2, \tag{24}$$

where $\Xi_R = [\varepsilon_1^{(R)}, \dots, \varepsilon_n^{(R)}]^\top$, $\varepsilon_i^{(R)} = r_t^i - h_R(s_t^i, a_t^i, w_t^i)$, and $K$ is the kernel matrix on $\{(s_t^i, a_t^i, z_t^i)\}_{i=1}^n$. Minimizing $\mathcal{L}_n^{(R)}(\theta)$ yields the desired reward-emission bridge function.

## H   MUJOCO BENCHMARKS

### H.1   ENVIRONMENT DETAILS

This section provides an overall overview of the environments for the four control tasks: Swimmer, Hopper, HalfCheetah, and Ant (See Figure 8). Table 2 summarizes the observation and action space specifications of the four environments. The transition dynamics of these MuJoCo environments are governed by the underlying physics simulator, which cannot be accessed or manipulated explicitly. Instead, the benchmark exposes a standard interface to query transitions: at each time step, the next state $s_{t+1}$ is obtained by calling the environment step function `env.step()` with the current action $a_t$.

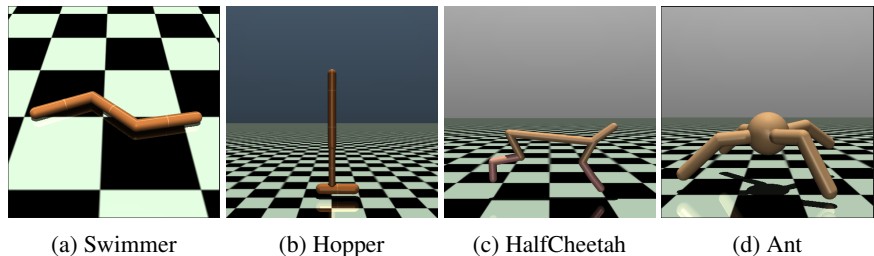

|          (a) Swimmer          |          (b) Hopper          |          (c) HalfCheetah          |          (d) Ant          |

Figure 8: Screenshots of four MuJoCo benchmark environments.

Table 2: Observation and action space specifications for the MuJoCo environments used in our experiments.

| Environment Name | Observation Space (Dim / Range / dtype) | Action Space (Dim / Range / dtype) |
|---|---|---|
| Swimmer | $8$ / $(-inf, inf)$ / float64 | $2$ / $[-1, 1]$ / float32 |
| Hopper | $11$ / $(-inf, inf)$ / float64 | $3$ / $[-1, 1]$ / float32 |
| HalfCheetah | $17$ / $(-inf, inf)$ / float64 | $6$ / $[-1, 1]$ / float32 |
| Ant | $105$ / $(-inf, inf)$ / float64 | $8$ / $[-1, 1]$ / float32 |

## H.2 DATA GENERATION PROCESS

Following (Shi et al., 2022; Miao et al., 2022), we generate synthetic data at each time step $t$ as follows. First, we draw an unobserved confounder $U_t \sim \mathcal{N}(0, I_{d_a})$ at each time step $t$. We then construct the action-inducing proxy $Z_t$ as a nonlinear transformation of $U_t$, $Z_t = \tanh(U_t) + \eta_t, \eta_t \sim \mathcal{N}(0, 0.05^2 I_{d_a})$, and form the outcome-inducing proxy $W_t = \tanh(U_t) + 0.2 U_t^2 + \varepsilon_t, \varepsilon_t \sim \mathcal{N}(0, 0.05^2 I_{d_a})$. The action is then set to $A_t = 0.5 Z_t + 0.5 U_t$ and clipped to the environment's action bounds. Executing $A_t$ in the environment yields the next state $S_{t+1}$ and reward $R_t$; the observed next state is obtained by adding the scalar $\sum_i U_{t,i}$ to every dimension of $S_{t+1}$, i.e., $S_{t+1} = S_{t+1} + \sum_i U_{t,i}$.

## H.3 ADDITIONAL EXPERIMENTAL RESULTS

In this section, we present additional experimental results. As shown in Table 3, our deconfounding algorithm can also be integrated into both the M2AC and MACURA frameworks. The original methods are unable to address the bias introduced by unobserved confounders. By incorporating our algorithm, these approaches gain the capability to mitigate such bias, thereby achieving more accurate and reliable estimation results.

## H.4 SENSITIVITY ANALYSIS

In this section, we present a series of sensitivity analyses. When parts of the proximal assumptions are violated, our method is able to retain a reasonable level of estimation accuracy, while the resulting bias increases progressively with the severity of the violation, exhibiting a deviation pattern consistent with theoretical expectations.

### H.4.1 COMPLEX CONFOUNDING ENVIRONMENT

To further assess the robustness of our approach under richer and more challenging confounding structures, we introduce an extended data-generation setting that incorporates highly nonlinear dependencies between the unobserved confounder and the observed variables. This construction follows recent developments in proximal causal inference Kompa et al. (2022).

**unobserved confounders.** At each time step, we sample an unobserved confounder $U_t \in \mathbb{R}^{d_a}$ from a multivariate normal distribution $N(0, I)$. The confounder is mapped through a smooth,

Table 3: Estimation error (mean $\pm$ std) across MuJoCo benchmark.

| Methods | Time steps (Rollout lengths) | Benchmark tasks | | | |
| --- | --- | --- | --- | --- | --- |
| | | Swimmer | Hopper | Half Cheetah | Ant |
| Confounded M2AC | 1 k ( 1-step) | $0.81 \pm 0.83$ | $1.55 \pm 1.74$ | $3.09 \pm 2.96$ | $8.17 \pm 5.02$ |
| | 10k ( 5-step) | $0.90 \pm 0.90$ | $1.36 \pm 1.50$ | $7.53 \pm 4.35$ | $9.18 \pm 5.40$ |
| | 20k (10-step) | $0.89 \pm 0.85$ | $1.62 \pm 1.61$ | $7.58 \pm 4.43$ | $7.61 \pm 4.72$ |
| | 30k (15-step) | $0.95 \pm 0.96$ | $1.81 \pm 1.45$ | $8.29 \pm 4.36$ | $8.28 \pm 5.01$ |
| DPC-M2AC | 1 k ( 1-step) | $0.11 \pm 0.08$ | $0.12 \pm 0.12$ | $0.90 \pm 0.48$ | $4.79 \pm 1.69$ |
| | 10k ( 5-step) | $0.13 \pm 0.09$ | $0.32 \pm 0.26$ | $5.82 \pm 3.07$ | $6.35 \pm 2.15$ |
| | 20k (10-step) | $0.15 \pm 0.09$ | $0.41 \pm 0.32$ | $5.97 \pm 3.01$ | $5.36 \pm 2.04$ |
| | 30k (15-step) | $0.21 \pm 0.14$ | $0.59 \pm 0.33$ | $6.82 \pm 3.66$ | $5.91 \pm 2.18$ |
| Confounded MACURA | 1 k ( 1-step) | $0.99 \pm 0.77$ | $1.65 \pm 1.95$ | $3.19 \pm 3.08$ | $7.43 \pm 5.42$ |
| | 10k ( 5-step) | $0.95 \pm 0.82$ | $0.83 \pm 1.00$ | $7.21 \pm 4.24$ | $8.38 \pm 3.62$ |
| | 20k (10-step) | $1.05 \pm 1.02$ | $1.67 \pm 1.74$ | $7.34 \pm 3.90$ | $9.03 \pm 6.16$ |
| | 30k (15-step) | $0.72 \pm 0.73$ | $1.49 \pm 1.50$ | $8.79 \pm 5.13$ | $8.48 \pm 4.72$ |
| DPC-MACURA | 1 k ( 1-step) | $0.10 \pm 0.07$ | $0.11 \pm 0.16$ | $0.77 \pm 0.45$ | $4.29 \pm 1.78$ |
| | 10k ( 5-step) | $0.13 \pm 0.09$ | $0.24 \pm 0.15$ | $5.38 \pm 2.83$ | $6.07 \pm 2.11$ |
| | 20k (10-step) | $0.14 \pm 0.11$ | $0.34 \pm 0.23$ | $5.63 \pm 2.60$ | $5.26 \pm 1.93$ |
| | 30k (15-step) | $0.14 \pm 0.09$ | $0.45 \pm 0.21$ | $6.11 \pm 3.24$ | $4.99 \pm 1.69$ |

nonlinear feature operator:

$$\psi(U_t) = \frac{(U_t - 0.5)^4}{40} + \exp\left(-2.5(U_t - 0.5)^2\right) + \frac{U_t - 0.5}{6}. \tag{25}$$

**Action-inducing proxy** $Z_t$. We generate $Z_t$ by combining $\psi(U_t)$ with harmonics of the confounder:

$$Z_t = 1.4 \sin(\Omega \psi(U_t) + \phi) + 0.9 \cos((\Omega + 0.3)U_t) + 0.25 \, \psi(U_t)^2$$
$$+ 0.1 \, U_t^3 + \varepsilon_t^{(Z)}, \qquad \varepsilon_t^{(Z)} \sim N(0, 0.08^2 I), \tag{26}$$

where the frequencies $\Omega$ are sampled from $[0.8, 1.6]$ and the phases $\phi$ from $[0, \pi/2]$.

**Outcome-inducing proxy** $W_t$. The outcome-inducing proxy is generated from another nonlinear transformation:

$$\psi_W(U_t) = \frac{U_t^4}{55} + \exp(-3U_t^2) + \frac{U_t}{5.5}, \tag{27}$$

$$W_t = 0.55 \, \psi_W(U_t) + 0.2 \, \psi_W(U_t)^2 + 0.85 \sin(1.1 U_t) + 0.6 \cos(0.9 \, \psi_W(U_t))$$
$$+ \varepsilon_t^{(W)}, \qquad \varepsilon_t^{(W)} \sim N(0, 0.07^2 I). \tag{28}$$

**Confounded next states.** To inject confounding into the next states, we add a confounder-induced scalar shift:

$$S_{t+1}^{\text{conf}} = S_{t+1} + \sum_i \psi(U_{t,i}), \tag{29}$$

where $S_{t+1}$ is the unconfounded next state provided by the MuJoCo environment. All other components of the experimental pipeline remain unchanged.

**Results.** This extended setting induces more complex data generation than the design used in Section H.2. As summarized in Table 4, across all environments, the model-based RL algorithms equipped with our deconfounding algorithm achieve lower MSEs than their confounded counterparts that ignore unobserved confounders. These results further validate the empirical benefits of our approach under complex confounding structures.

Table 4: Model estimation errors (mean ± std) across MuJoCo benchmarks under the complex confounding structure.

| Methods | Time steps (Rollout lengths) | Benchmark tasks | | | |
|---|---|---|---|---|---|
| | | Swimmer | Hopper | Half Cheetah | Ant |
| Confounded MBPO | 1k (1-step) | $1.73 \pm 1.93$ | $3.08 \pm 2.85$ | $16.56 \pm 9.52$ | $38.80 \pm 25.39$ |
| | 10k (5-step) | $2.35 \pm 1.77$ | $3.50 \pm 3.82$ | $23.96 \pm 15.93$ | $43.46 \pm 20.39$ |
| | 20k (10-step) | $2.66 \pm 1.23$ | $4.83 \pm 4.21$ | $27.31 \pm 10.05$ | $53.98 \pm 26.41$ |
| | 30k (15-step) | $2.93 \pm 1.89$ | $7.72 \pm 5.46$ | $43.99 \pm 23.98$ | $91.06 \pm 35.81$ |
| DPC-MBPO | 1k (1-step) | $0.73 \pm 0.48$ | $0.47 \pm 0.56$ | $5.10 \pm 1.96$ | $9.00 \pm 4.33$ |
| | 10k (5-step) | $1.22 \pm 0.78$ | $0.64 \pm 0.40$ | $12.13 \pm 6.08$ | $14.61 \pm 5.15$ |
| | 20k (10-step) | $1.50 \pm 0.72$ | $1.27 \pm 0.46$ | $14.98 \pm 6.92$ | $16.86 \pm 7.67$ |
| | 30k (15-step) | $2.49 \pm 0.93$ | $2.65 \pm 1.47$ | $17.52 \pm 7.88$ | $25.57 \pm 8.37$ |
| Confounded M2AC | 1k (1-step) | $1.62 \pm 1.78$ | $2.95 \pm 3.21$ | $11.36 \pm 6.85$ | $27.11 \pm 15.78$ |
| | 10k (5-step) | $2.41 \pm 2.21$ | $3.77 \pm 3.73$ | $19.55 \pm 11.70$ | $27.43 \pm 15.06$ |
| | 20k (10-step) | $2.30 \pm 1.95$ | $3.47 \pm 3.64$ | $20.17 \pm 12.50$ | $25.71 \pm 16.72$ |
| | 30k (15-step) | $2.47 \pm 2.05$ | $3.87 \pm 3.72$ | $22.74 \pm 11.22$ | $23.25 \pm 11.31$ |
| DPC-M2AC | 1k (1-step) | $0.40 \pm 0.30$ | $0.28 \pm 0.31$ | $3.31 \pm 1.29$ | $6.44 \pm 2.48$ |
| | 10k (5-step) | $0.53 \pm 0.43$ | $0.46 \pm 0.31$ | $6.84 \pm 3.84$ | $8.19 \pm 2.51$ |
| | 20k (10-step) | $0.55 \pm 0.37$ | $0.58 \pm 0.41$ | $7.74 \pm 4.77$ | $7.78 \pm 3.24$ |
| | 30k (15-step) | $0.90 \pm 0.70$ | $0.85 \pm 0.55$ | $11.04 \pm 5.42$ | $9.08 \pm 3.10$ |
| Confounded MACURA | 1k (1-step) | $1.78 \pm 1.51$ | $2.81 \pm 3.46$ | $13.90 \pm 9.94$ | $26.29 \pm 18.88$ |
| | 10k (5-step) | $1.93 \pm 1.81$ | $3.45 \pm 3.69$ | $18.96 \pm 10.41$ | $27.31 \pm 17.95$ |
| | 20k (10-step) | $2.19 \pm 1.65$ | $3.55 \pm 4.30$ | $18.39 \pm 10.91$ | $27.40 \pm 16.02$ |
| | 30k (15-step) | $2.26 \pm 2.26$ | $4.91 \pm 5.00$ | $18.20 \pm 9.30$ | $27.34 \pm 13.25$ |
| DPC-MACURA | 1k (1-step) | $0.42 \pm 0.35$ | $0.47 \pm 0.43$ | $3.54 \pm 1.58$ | $7.66 \pm 3.49$ |
| | 10k (5-step) | $0.83 \pm 0.60$ | $0.64 \pm 0.55$ | $8.69 \pm 4.33$ | $8.95 \pm 3.59$ |
| | 20k (10-step) | $0.84 \pm 0.63$ | $0.96 \pm 0.77$ | $9.59 \pm 5.34$ | $8.95 \pm 3.59$ |
| | 30k (15-step) | $0.98 \pm 0.56$ | $1.76 \pm 1.12$ | $10.16 \pm 6.31$ | $8.76 \pm 3.20$ |

### H.4.2 EXCLUSION ASSUMPTION VIOLATION

To present the performance of our approach when the proximal assumptions are violated, we design an experimental setting that explicitly breaks the exclusion restrictions. We retain the baseline construction of the unobserved confounder $U_t$ and the action-inducing proxy $Z_t$ in Section H.2, but introduce two additional structural pathways.

First, we inject a direct causal link from the action to the outcome-inducing proxy $W_t$ by modifying its generation as

$$W_t \leftarrow W_t + 0.1A_t,$$

thereby establishing an explicit violation of the exclusion restriction $A_t \not\rightarrow W_t$. Second, we introduce a dependence of both the next state on the action-inducing proxy $Z_t$ by:

$$S_{t+1} \leftarrow S_{t+1} + 0.5 \sum_i Z_{t,i},$$

creating the paths $Z_t \rightarrow S_{t+1}$.

Table 5 summarizes the empirical performance when the Exclusion Assumption is violated. As expected, violating the assumptions leads to higher estimation errors compared to the setting where the assumptions hold (see Table 1). Interestingly, however, we observe a stable and gradual performance deterioration rather than catastrophic failure. This pattern suggests that the method retains a degree of robustness under modest violations of the proximal assumptions, a phenomenon also observed in recent proximal causal studies Huang & McCartan (2025). Quantifying such robustness formally, characterizing how bias scales with the extent of assumption violation, represents an important direction for future research.

Table 5: Estimation error (mean $\pm$ std) under Exclusion assumption violations across MuJoCo benchmarks.

| Methods | Time steps (Rollout lengths) | Benchmark tasks | | | |
|---|---|---|---|---|---|
| | | Swimmer | Hopper | Half Cheetah | Ant |
| DPC-MBPO | 1k (1-step) | $0.30 \pm 0.22$ | $0.31 \pm 0.25$ | $2.06 \pm 1.06$ | $5.54 \pm 2.05$ |
| | 10k (5-step) | $0.36 \pm 0.18$ | $0.44 \pm 0.26$ | $7.50 \pm 5.42$ | $6.55 \pm 2.56$ |
| | 20k (10-step) | $0.53 \pm 0.29$ | $0.72 \pm 0.41$ | $8.16 \pm 4.39$ | $7.94 \pm 2.87$ |
| | 30k (15-step) | $0.98 \pm 0.83$ | $1.45 \pm 0.86$ | $9.99 \pm 5.66$ | $13.26 \pm 4.94$ |
| DPC-M2AC | 1k (1-step) | $0.14 \pm 0.10$ | $0.17 \pm 0.15$ | $0.98 \pm 0.48$ | $4.79 \pm 1.61$ |
| | 10k (5-step) | $0.16 \pm 0.11$ | $0.40 \pm 0.29$ | $6.32 \pm 3.25$ | $6.58 \pm 2.06$ |
| | 20k (10-step) | $0.20 \pm 0.17$ | $0.50 \pm 0.40$ | $6.43 \pm 3.53$ | $6.07 \pm 2.02$ |
| | 30k (15-step) | $0.31 \pm 0.20$ | $0.80 \pm 0.63$ | $7.50 \pm 3.57$ | $6.42 \pm 2.32$ |
| DPC-MACURA | 1k (1-step) | $0.12 \pm 0.13$ | $0.18 \pm 0.13$ | $2.98 \pm 1.56$ | $4.78 \pm 1.54$ |
| | 10k (5-step) | $0.13 \pm 0.13$ | $0.31 \pm 0.27$ | $5.58 \pm 3.02$ | $5.88 \pm 2.34$ |
| | 20k (10-step) | $0.15 \pm 0.10$ | $0.39 \pm 0.21$ | $6.10 \pm 3.38$ | $5.38 \pm 1.67$ |
| | 30k (15-step) | $0.24 \pm 0.37$ | $0.44 \pm 0.29$ | $6.38 \pm 3.27$ | $5.23 \pm 1.30$ |

### H.4.3 COMPLETENESS ASSUMPTION VIOLATION

To present the performance of our approach when the Completeness assumptions are violated, we further design an experiment in which the assumption is intentionally violated. In this setting, we retain the baseline construction of the unobserved confounder $U_t$ and the action-inducing proxy $Z_t$, but replace the outcome-inducing proxy $W_t$ with an almost constant signal:

$$W_t = c + \varepsilon_t, \qquad \varepsilon_t \sim N(0, \sigma^2 I),$$

where $c$ is a fixed vector (each entry set to 0.1) and $\sigma = 0.05$. This modification removes meaningful variation in $W_t$ that would otherwise carry information about the confounder $U_t$, thereby directly violating the completeness assumptions. Table 6 summarizes the model estimation errors under this regime. Across all MuJoCo environments, we observe an increase in MSE compared with the original setting where completeness holds (see Table 1).

Table 6: Estimation error (mean $\pm$ std) under completeness assumption violations across MuJoCo benchmarks.

| Methods | Time steps (Rollout lengths) | Benchmark tasks | | | |
|---|---|---|---|---|---|
| | | Swimmer | Hopper | Half Cheetah | Ant |
| DPC-MBPO | 1k (1-step) | $0.64 \pm 0.63$ | $1.05 \pm 1.04$ | $3.01 \pm 2.83$ | $8.42 \pm 5.09$ |
| | 10k (5-step) | $0.69 \pm 0.69$ | $0.95 \pm 0.82$ | $8.31 \pm 4.40$ | $9.26 \pm 4.57$ |
| | 20k (10-step) | $0.85 \pm 0.67$ | $1.85 \pm 1.89$ | $10.18 \pm 4.96$ | $9.68 \pm 4.17$ |
| | 30k (15-step) | $1.17 \pm 0.97$ | $2.37 \pm 1.33$ | $11.15 \pm 5.14$ | $11.98 \pm 4.38$ |
| DPC-M2AC | 1k (1-step) | $0.61 \pm 0.58$ | $0.89 \pm 1.11$ | $2.54 \pm 2.68$ | $8.97 \pm 6.50$ |
| | 10k (5-step) | $0.79 \pm 0.73$ | $1.06 \pm 0.97$ | $7.38 \pm 4.80$ | $8.49 \pm 4.23$ |
| | 20k (10-step) | $0.63 \pm 0.64$ | $1.72 \pm 1.47$ | $8.49 \pm 4.46$ | $9.45 \pm 5.24$ |
| | 30k (15-step) | $0.75 \pm 0.72$ | $1.60 \pm 1.38$ | $8.97 \pm 4.72$ | $8.77 \pm 5.80$ |
| DPC-MACURA | 1k (1-step) | $0.71 \pm 0.69$ | $1.11 \pm 1.23$ | $2.68 \pm 2.83$ | $8.19 \pm 5.55$ |
| | 10k (5-step) | $0.81 \pm 0.71$ | $1.10 \pm 1.14$ | $6.80 \pm 3.78$ | $8.32 \pm 5.73$ |
| | 20k (10-step) | $0.72 \pm 0.64$ | $1.41 \pm 1.40$ | $9.37 \pm 5.49$ | $7.45 \pm 4.53$ |
| | 30k (15-step) | $0.71 \pm 0.58$ | $1.49 \pm 1.32$ | $9.60 \pm 7.59$ | $7.27 \pm 3.35$ |

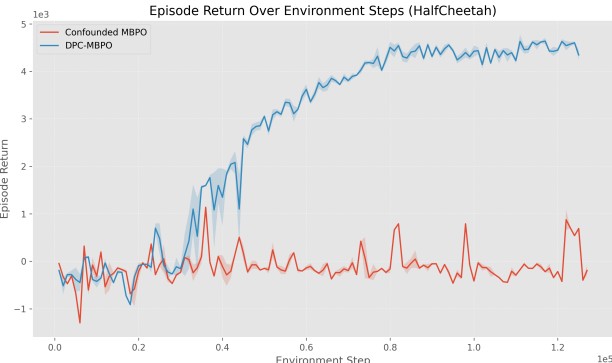

Figure 9: Policy learning performance in HalfCheetah. The solid lines denote the mean return over five evaluation episodes, and the shaded areas represent the standard deviation.

### H.4.4 POLICY LEARNING DEMONSTRATION

To examine the overall performance of the learned policy after completing the full procedure in Algorithm 1, we conduct a policy learning demonstration in the HalfCheetah environment. We follow the same experimental setup as in the model-estimation experiments, and introduce confounding in the reward function through the following mechanism:

$$r_t^{\text{conf}} = r_t + 0.3 \sum_i U_{t,i},$$

where $U_t$ denotes the unobserved confounder.

Figure 9 presents the learning curves, where the solid lines show the average return over five evaluation episodes and the shaded regions represent one standard deviation. The results indicate that DPC-MBPO is able to learn a high-performing policy even in the presence of unobserved confounders. While a performance gap remains relative to the oracle optimal policy Janner et al. (2019), the learned policy outperforms the confounded MBPO baseline, which fails to achieve meaningful improvement throughout training. This provides preliminary evidence that completing the full proximal model-based RL procedure indeed improves downstream policy learning performance.

# I   RHC EXPERIMENTS

## I.1   DESCRIPTION OF THE RHC DATASET

The dataset consists of treatment records from 5,735 patients. Each patient's state $s$ is represented by 71 covariates, encompassing demographic characteristics, diagnostic information, comorbidities, vital signs, physiological status, and functional status, consisting of binary, categorical, and continuous variables. At the treatment decision stage, the clinician chooses an action $a \in \{0, 1\}$, where $a = 1$ corresponds to administering RHC and $a = 0$ corresponds to withholding it. Following these decisions, each trajectory terminates with a clinical outcome signal $r$, defined as the time from admission to either death or censoring within 30 days. Further discussion on variable specification can be found in Hirano & Imbens (2001). Importantly, potential confounding in this dataset may arise from the fact that several laboratory-based physiological measurements collected at admission are subject to substantial measurement error. In addition, beyond such errors in laboratory evaluations, the possibility of additional unobserved confounders can influence both treatment assignment and outcomes, including lifestyle-related factors (such as smoking history, dietary habits, and physical activity) and socioeconomic conditions (such as income level, education, and access to healthcare).

In this experiment, what we referred to as "ground truth" corresponds to a reference value from the factual 30-day survival outcomes under the clinician policy[12]. Specifically, we compute the average 30-day survival outcome under the clinician policy and use this value as a comparison point for evaluating the relative consistency of different OPE methods. The log-MSE metric measures deviation from this reference baseline, not error to a true counterfactual value.

## I.2   SELECTION OF PROXY VARIABLES

In the SUPPORT experiments, proxy variables were selected from an initial pool of 71 candidate covariates following established procedures in proximal causal inference Shen & Cui (2023). For each variable, we assessed its association with the treatment $A$ and the 30-day survival outcome $Y$ using Spearman correlation tests for continuous variables and chi-squared tests for categorical variables, producing the screening statistics $p_{\text{val\_with\_A}}$ and $p_{\text{val\_with\_Y}}$. Action-inducing proxies $Z_t$ were chosen as variables that exhibited strong association with the treatment but only weak association with the outcome, while outcome-inducing proxies $W_t$ were selected analogously in the reverse direction. To ensure robustness, we further ranked all variables according to the pair $(p_{\text{val\_with\_A}}, p_{\text{val\_with\_Y}})$ and evaluated multiple proxy-set configurations of increasing size, enabling the construction of progressively more "sufficiently rich" proxy sets required for proximal identification.

Based on the ranking derived from the association measures $p_{\text{val\_with\_A}}$ and $p_{\text{val\_with\_Y}}$, we identified four action-inducing proxies and four outcome-inducing proxies. The selected action-inducing proxies include *pafi1* (PaO$_2$/FiO$_2$ ratio at admission), *paco21* (arterial partial pressure of CO$_2$), *wtkilol* (body weight), and *hrt1* (admission heart rate). The corresponding outcome-inducing proxies consist of *ph1* (arterial blood pH), *hemal* (hematocrit level), *ca* (cancer severity indicator), and *age* (patient age).

To evaluate robustness with respect to the different proxy set, we constructed multiple experimental configurations by selecting the top 2, 4, 6, and 8 proxies according to their ranking. These progressively larger proxy sets enable examination of how identification quality and estimation performance vary as the proxy information becomes increasingly "sufficiently rich" for proximal causal learning. The more detailed strategy for proxy selection can be found in Shen & Cui (2023).

## I.3   RUNTIME AND COMPUTATIONAL COST

The experiments were conducted on the National Computational Infrastructure (NCI Australia) under same hardware configuration. Table 7 reports the wall-clock runtime of the main baselines and our proposed method.

The additional runtime of our approach primarily arises from the proximal causal components, which require solving kernel-based moment conditions. These computations involve Gaussian kernels constructed using pairwise distances and the combination of multiple kernel matrices, leading to inherent $O(n^2)$–$O(n^3)$ memory and computational overhead. As a result, our method is slower than

Table 7: Runtime of different methods under the same hardware configuration.

| Method | Runtime |
|---|---|
| MBPO (vanilla) | $\sim$10 minutes |
| pFOE | $\sim$40 minutes |
| PCI | $\sim$1 hour |
| DPC-MBRL (ours) | $\sim$1.2 hours |

purely feed-forward model-based RL baselines such as MBPO. In addition, the model-free proximal baselines pFOE and PCI also incur higher computational cost compared to vanilla MBPO. Both rely on doubly robust estimators and require alternating updates between value and policy objectives, coupled with regularization terms that must be optimized jointly, which increases their end-to-end wall-clock time.

