# OpenReview forum: "Trust the model when it is confounded: Model-based Reinforcement learning for Confounded POMDPs"
_ICLR.cc/2026/Conference — Submitted to ICLR 2026_

### Official Review · Reviewer_ago2 · 2025-10-17

**Soundness:** 2
**Presentation:** 3
**Contribution:** 2
**Rating:** 4
**Confidence:** 4

**Summary:**

This paper addresses the challenge of learning dynamics models for model based RL in environments with unobserved confounders. The authors propose a method which reformulates the dynamics using proximal causal inference. The authors use neural networks to model bridge functions a Maximum Moment Restriction framework to train them. The authors validate their method on two benchmarks: a mujoco environment with synthetically added confounders and the real-world SUPPORT dataset. The results show that the method leads to more accurate dynamics models and superior policy value estimation compared to standard model based and model-free methods that do not account for confounding.

**Strengths:**

1. The paper bridges the gap between the proximal causal inference literature and model-based RL. Using the bridge function $h_D$ as a de-biased dynamics model is an original idea.
2. The method in the paper scales to continuous and high-dimensional spaces, overcoming the limitations of previous tabular and kernel-based methods.
3. The authors do an excellent job in clearly formalizing the problem of confounding bias inthe dynamics, contrasting the observational expectation with the required interventional expectation.
4. The experiments demonstrate a benefit of the proposed approach. They also show that if the proximal assumptions are met or assumed to be met, the method effectively reduces bias and improves performance.

**Weaknesses:**

1. Related to the practical implications of this work, the work on "Delphic Offline Reinforcement Learning" (Pace et al. 2023) addresses a very similar problem (i.e., offline RL with hidden confounders) in a similar application domain (medical EHRs). I find it critical for this work to position itself relative to that. I would consider increasing my score if the authors add empirical comparisons.
2. The entire method's validity rests on unverifiable assumptions of proximal causal inference: the existence of "sufficiently rich" proxies ($W_t, Z_t$) that precisely match the causal DAG in Figure 2. The paper itself notes this is "non-trivial" and a limitation. This emphasizes again the importance I find to comparing to the practical method in weakness 1, as it would demonstrate the significance of these assumptions. \
\
This reliance on strong assumptions is is also evident by the experimental design. The mujoco experiment is a "best-case" tautology, as the proxies are synthetically generated to perfectly fit the model's requirements, which does not test robustness. The SUPPORT experiments, which is the most compelling, omit the most critical detail: the selection and justification of the proxies. The authors should not defer this to an external citation (i.e., Shen & Cui, 2023).

3. The presentation of Section 3.2 ("Estimation of Bridge Functions") implies a novel methodological development. In fact, it is a direct and faithful application of the NMMR method (Kompa et al., 2022)  to the dynamics/reward functions. The novelty is in the application, not the method.

**Questions:**

1. In the SUPPORT experiment, which of the covariates were selected as $Z_t$ and $W_t$? What is the domain-knowledge justification that these variables satisfy the causal assumptions? (e.g., $Z_t$ only affects $A_t$ via $U_t$; $W_t$ is not affected by $A_t$; etc.). Why should we believe these proxies are "sufficiently rich" to satisfy the completeness assumption? The real-world results are unconvincing without this justification.
2. How does your method perform when the proximal assumptions are violated? For example, in the mujoco setting, what happens to the model's MSE if you introduce a weak causal link from $Z_t$ directly to $S_{t+1}$ or from $A_t$ directly to $W_t$?
3. How was the "ground truth" policy value determined in Figure 3 for the SUPPORT dataset? This is a real-world offline dataset, so there is no access to a true environment or ground truth. Please clarify what this (log-mse) metric is being computed against.

---

> ### Author Response · Authors · 2025-11-27
>
> #### We sincerely appreciate your detailed comments and your recognition of our work. We provide our responses to your questions below.
>
> #### **Weaknesses 1**: "Related to the practical implications of this work, the work on "Delphic Offline Reinforcement Learning" (Pace et al. 2023) addresses a very similar problem (i.e., offline RL with hidden confounders) in a similar application domain (medical EHRs). I find it critical for this work to position itself relative to that. I would consider increasing my score if the authors add empirical comparisons."
>
> > **Response**: We thank the reviewer for referring us to this related work [1] (i.e. Pace et al. 2023). Our work and this work both address unobserved confounding in data by leveraging causal inference. However, the two approaches rely on fundamentally different causal inference techniques.
>
> >Our approach is grounded in the direct use of auxiliary variables which meet certain assumptions to identify causal effects without trying to recover the unobserved confounder U. In contrast, the work by (Pace et al. 2023) seeks to recover or approximate the unobserved confounder U via representation learning, then standard confounding adjustment methods are used to eliminate the confounding bias.
>
> >Both categories of deconfounding methods are well-established in causal inference and have been applied in sequential decision-making. Each offers distinct advantages and trade-offs. The auxiliary variable approach provides explicit identification guarantees but relies on the availability of sufficiently rich auxiliary variables, whereas the latent representation learning approach, while may appear more flexible and applicable in practice, it lacks identifiability guarantee since the success of the approach depends on model specification and whether the data contains sufficient information for recovering the confounder [2].
>
> >We really hope we can conduct an empirical comparison of the two approaches, but as the preprocessed dataset used in the original implementation of the method in [1] is not publicly available, we have not been able to reproduce the results to date. We will continue to explore possible ways to do so.
>
> #### **Weaknesses 2**: "The entire method's validity rests on unverifiable assumptions of proximal causal inference: the existence of "sufficiently rich" proxies ($W_t,Z_t$) that precisely match the causal DAG in Figure 2. The paper itself notes this is "non-trivial" and a limitation. This emphasizes again the importance I find to comparing to the practical method in weakness 1, as it would demonstrate the significance of these assumptions. This reliance on strong assumptions is is also evident by the experimental design. The mujoco experiment is a "best-case" tautology, as the proxies are synthetically generated to perfectly fit the model's requirements, which does not test robustness. The SUPPORT experiments, which is the most compelling, omit the most critical detail: the selection and justification of the proxies. The authors should not defer this to an external citation (i.e., Shen & Cui, 2023)."
>
> > **Response**: We sincerely thank the reviewer for raising this important concern.
>
> >(1)	We agree that in general obtaining valid auxiliary variables such as the proxies required in our paper or instrumental variables is not always straightforward in real environments. Fortunately, in many cases, domain knowledge can provide sufficient guidance on this, and some applied studies, e.g., [3-4] have drawn on domain expertise to identify auxiliary variables that meet relevant assumptions, thereby enabling deconfounding. They also provide practical, scenario-specific guidance on how to select them. Here, we provide two examples:
>
> >First, [3] shows that insurance plans can serve as an auxiliary variable for a patient’s treatment assignment, as it influences the treatment choice but is independent of the clinical unmeasured confounders. Second, [5] uses the patient’s oxygenation status as an action-inducing proxy for whether clinicians decide to perform right-heart catheterization (RHC), given its strong correlation with the intubation decision. Then hematocrit is used as an outcome-inducing proxy because it is associated with the eventual clinical outcome.
>
> >In addition, in scenarios where expertise knowledge is limited, it would be possible to learn the proxies from data directly. Learning the representations of auxiliary variables, especially instrumental variables from data is an emerging research topic in the causal inference field [6-7]. Following this line of work, our approach can also be extended by incorporating representation learning to automatically construct the proxy variables from raw observations, which will be a very promising future direction for us. We have mentioned this challenge and outlined this future research direction in the Conclusion section of the paper.

---

> ### Author Response · Authors · 2025-11-27
>
> > Response to Weakness 2 (continued):
>
> >(2)	Regarding the selection of proxy variables in the SUPPORT experiment, we agree with the reviewer that the current draft omits some details. We refer the reviewer to our detailed clarification provided in response to Question 1, and we have added this description of how the proxies were chosen to Appendix I.2 of the revised paper.
>
> #### **Weaknesses 3**: "The presentation of Section 3.2 ("Estimation of Bridge Functions") implies a novel methodological development. In fact, it is a direct and faithful application of the NMMR method (Kompa et al., 2022) to the dynamics/reward functions. The novelty is in the application, not the method."
>
> > **Response**: We thank the reviewer for this insightful comment. We agree that the estimation of bridge functions itself is not a novel methodological development. Our primary contribution is to leverage proxy variables and bridge functions for learning both the dynamics/reward functions to tackle the challenge of unobserved confounding in model-based RL. In the revised version, we will make this point more explicit.
>
> #### **Question 1**: "In the SUPPORT experiment, which of the covariates were selected as $Z_t$ and $W_t$? What is the domain-knowledge justification that these variables satisfy the causal assumptions? (e.g., $Z_t$ only affects $A_t$ via $U_t$; $W_t$ is not affected by $A_t$; etc.). Why should we believe these proxies are "sufficiently rich" to satisfy the completeness assumption? The real-world results are unconvincing without this justification."
>
> > **Response**: We thank the reviewer for pointing out the need for greater clarity regarding how proxy variables were selected in the SUPPORT experiments. To identify suitable proxies, we began with 71 candidate variables in the SUPPORT dataset and followed the following procedure in the prior work [5]:
>
> >1) Association with treatment and outcome. For each variable, we tested its association with treatment A and 30-day survival outcome Y using Spearman tests for continuous variables and χ² tests for categorical variables. The resulting p-values were recorded as p_val_with_A and p_val_with_Y.
> >2) Action-inducing proxies Z_t should be strongly associated with A but not directly with Y. Therefore, we prioritized variables Z_t with small p_val_with_A and large p_val_with_Y. Outcome-inducing proxies W_t were selected analogously but in reverse.
> >3) To ensure robustness, we further ranked variables using a combined score derived from the pair (p_val_with_A, p_val_with_Y) and evaluated multiple proxy configurations (e.g., one pair proxy to four pairs proxies) to check an increasingly large collection of proxies, thereby making the proxy set more “sufficiently rich”.
>
> >Based on the ranking, we identified four action-inducing proxies and four outcome-inducing proxies. The selected action-inducing proxies include *pafi1* (PaO₂/FiO₂ ratio at admission), *paco21* (arterial partial pressure of CO₂), *wtkilol* (body weight), and *hrt1* (admission heart rate). The corresponding outcome-inducing proxies consist of *ph1* (arterial blood pH), *hemal* (hematocrit level), *ca* (cancer severity indicator), and *age* (patient age).
>
> >We have added this description of how the proxies were chosen to Appendix I.2 of the revised paper.

---

> ### Author Response · Authors · 2025-11-27
>
> #### **Question 2**: "How does your method perform when the proximal assumptions are violated? For example, in the mujoco setting, what happens to the model's MSE if you introduce a weak causal link from $Z_t$ directly to $S_{t+1}$ or from $A_t$ directly to $W_t$?"
>
> > **Response**: Thank you for your question. To evaluate how our method behaves when the relevant assumptions are violated, following the comment, we have conducted supplementary experiments that explicitly break the exclusion restrictions underlying proximal causal learning. Specifically, we retain the baseline construction of the confounder $U_t$ and action-inducing proxy $Z_t$, but allow the action to feed directly into the outcome-inducing proxy $W_t$: $W_t = W_t + 0.1A_t$. We further inject the $Z_t$ into the environment by adding $0.5 \sum_i Z_{t,i}$ to the next state. These modifications create explicit structural paths $A_t \rightarrow W_t$ and $Z_t \rightarrow S_{t+1}$, thereby breaking the exclusion restriction.
>
> >The empirical results are summarized in the table below. Overall, once the proximal assumption is violated, the estimation error of the learned dynamics model becomes noticeably higher compared to the setting where the assumptions hold (See Table 1 in the original text). This deterioration is expected: violation of the proximal conditions breaks the underlying moment equations that identify the bridge functions, which in turn propagates bias into both the transition and reward models. Importantly, however, we still observe a stable degradation pattern rather than catastrophic failure, suggesting that the method retains a degree of robustness even under modest assumption violations, which is a property also noted in recent proximal-causal studies [8]. We believe this also highlights an interesting future direction, namely, to formally quantify how causal identification strategies degrade, in terms of bias or estimation error, as their underlying assumptions are progressively violated.
> | Methods       | Timesteps     | Swimmer       | Hopper        | Half Cheetah   | Ant           |
> |---------------|---------------|---------------|---------------|----------------|---------------|
> | DPC-MBPO      | 1k (1-step)   | 0.30±0.22     | 0.31±0.25     | 2.06±1.06      | 5.54±2.05     |
> |               | 10k (5-step)  | 0.36±0.18     | 0.44±0.26     | 7.50±5.42      | 6.55±2.56     |
> |               | 20k (10-step) | 0.53±0.29     | 0.72±0.41     | 8.16±4.39      | 7.94±2.87     |
> |               | 30k (15-step) | 0.98±0.83     | 1.45±0.86     | 9.99±5.66      | 13.26±4.94    |
> | DPC-M2AC      | 1k (1-step)   | 0.14±0.10     | 0.17±0.15     | 0.98±0.48      | 4.79±1.61     |
> |               | 10k (5-step)  | 0.16±0.11     | 0.40±0.29     | 6.32±3.25      | 6.58±2.06     |
> |               | 20k (10-step) | 0.20±0.17     | 0.50±0.40     | 6.43±3.53      | 6.07±2.02     |
> |               | 30k (15-step) | 0.31±0.20     | 0.80±0.63     | 7.50±3.57      | 6.42±2.32     |
> | DPC-MACURA    | 1k (1-step)   | 0.12±0.13     | 0.18±0.13     | 2.98±1.56      | 4.78±1.54     |
> |               | 10k (5-step)  | 0.13±0.13     | 0.31±0.27     | 5.58±3.02      | 5.88±2.34     |
> |               | 20k (10-step) | 0.15±0.10     | 0.39±0.21     | 6.10±3.38      | 5.38±1.67     |
> |               | 30k (15-step) | 0.24±0.37     | 0.44±0.29     | 6.38±3.27      | 5.23±1.30     |
>
>
> >We have added the extra experiments to Appendix H.4.2 of the revised paper.

---

> ### Author Response · Authors · 2025-11-27
>
> #### **Questions 3**: "How was the "ground truth" policy value determined in Figure 3 for the SUPPORT dataset? This is a real-world offline dataset, so there is no access to a true environment or ground truth. Please clarify what this (log-mse) metric is being computed against."
>
> > **Response**: Thank you for carefully examining this point. We agree that our original wording was inaccurate, especially in a clinical setting where no “ground truth” outcomes are available.
>
> >In this experiment, what we referred to as “ground truth” corresponds to a reference value from the factual 30-day survival outcomes under the clinician policy in the SUPPORT dataset [5]. Specifically, we compute the average 30-day survival outcome under the clinician policy and use this value as a comparison point for evaluating the relative consistency of different OPE methods. The log-MSE metric measures deviation from this reference baseline, not error to a true counterfactual value.
>
> >We have updated the paper to clarify this in Appendix I.1.
>
> #### **References**:
> >[1] Pace A, Yèche H, Schölkopf B, Ratsch G, Tennenholtz G. "Delphic Offline Reinforcement Learning under Nonidentifiable Hidden Confounding." ICLR.
> [2] Schölkopf B, Locatello F, Bauer S, Ke NR, Kalchbrenner N, Goyal A, Bengio Y. "Toward causal representation learning." Proceedings of the IEEE, 2021.
> [3] Ertefaie A, Small DS, Flory JH, Hennessy S. "A tutorial on the use of instrumental variables in pharmacoepidemiology." Pharmacoepidemiology and Drug Safety, 2017.
> [4] Brookhart MA, Rassen JA, Schneeweiss S. "Instrumental variable methods in comparative safety and effectiveness research." Pharmacoepidemiology and Drug Safety, 2010.
> [5] Shen T, Cui Y. "Optimal treatment regimes for proximal causal learning." NeurIPS, 2023.
> [6] Xu H, Xu Y, Li C, Zhuang F. "Causal structure representation learning of unobserved confounders in latent space for recommendation." ACM Transactions on Information Systems, 2025.
> [7] Cheng D, Xu Z, Li J, Liu L, Le TD, Liu J. "Learning conditional instrumental variable representation for causal effect estimation." ECML/PKDD, 2023.
> [8] Huang M, McCartan C. "Relative bias under imperfect identification in observational causal inference." arXiv, 2025.

---

### Official Review · Reviewer_dwrX · 2025-10-27

**Soundness:** 3
**Presentation:** 3
**Contribution:** 3
**Rating:** 6
**Confidence:** 4

**Summary:**

This work focuses on learning a dynamics model (and subsequently a policy) in POMDPs with unobservable confounders. They assume two sets of proxy variables, $Z_t, W_t$, along with corresponding bridge functions that connect the effects of the unobserved confounders $U_t$ to the observed variables (e.g., states) and the proxies. The bridge functions are learned using deep neural networks, and the empirical advantages of the proposed method are demonstrated on MuJoCo simulators and a clinical dataset.

**Strengths:**

- Clarity & Quality: The challenges of model learning in confounded POMDPs, arising from unobserved confounders that cause bias in the estimates, are clearly explained in Section 2.2. The assumptions required to enable the estimation of bridge functions are also well described. Algorithm 1 clearly illustrates how the bridge functions can be estimated from rollouts in the real environment and how a policy can be learned by rolling out imagined trajectories under the learned bridge functions.
- Significance: Overall, this work tackles a realistic and relevant problem in RL, particularly for high-dimensional states, since most data collection and learning occur in POMDPs rather than MDPs, and it is infeasible to assume that the states fully encode all information about the environment.
- Originality: This work adopts the bridge functions used in model-free POMDPs (e.g., Tchetgen Tchetgen et al., 2024) for model-based RL. While a two-stage estimation procedure is standard in MBRL, this approach improves on existing MBRL methods for POMDPs by removing the assumption by Grasse et al., 20223 that the underlying states are tabular and the dependence on kernel functions by Hong et al., 2024.

**Weaknesses:**

- MuJoCo domains, where the unobserved confounder is nonlinearly transformed to create $Z_t, W_t$ (described in H.2) are too simple to make convincing claims about the empirical advantages of this algorithm.
- Algorithm 1 describes a two-step process for policy learning. However, the experiments either report the error of the learned dynamics model or the OPE estimation errors, and do not present the overall performance of the learned policy resulting from completing the full procedure in Algorithm 1. While theoretically one might expect that reducing dynamics or OPE estimation errors could lead to improved policy learning, it would strengthen the work to compare the obtained policy’s value against that of the baselines or the oracle (if the oracle optimal policy is known).
- Another key limitation of this work, as the authors also acknowledge, is its assumption about the existence of valid proxy variables that satisfy Assumption 3.2. In realistic scenarios, obtaining proxy variables $W_t, Z_t$ may not be feasible. This also speaks to my concern raised earlier that the MuJoCo environments and the data-generating process (i.e., transforming $U_t$ to $Z_t$ and $W_t$ via a tanh function and adding a very small Gaussian noise) are too simplistic to effectively model real-world systems.

**Questions:**

For both experiments (Table 1 and the box plot in Figure 3), it would be beneficial to present the overall performance of the learned policy obtained after completing the full procedure described in Algorithm 1. This would also allow the authors to compare their results with model-free baselines in POMDPs with unobservable confounders (e.g., Tchetgen Tchetgen et al., 2024; Miao et al., 2024), as the current Table 1 only compares against the confounded model-based RL setting ("Confounded MBPO") and lacks comparisons with other baselines that would help contextualize the strength of the proposed method.

---

> ### Author Response · Authors · 2025-11-27
>
> #### We thank you for your thoughtful feedback. We appreciate your acknowledgment of the significance of our work and have addressed your concerns in the sequel.
>
> #### **Weaknesses 1**: "MuJoCo domains, where the unobserved confounder is nonlinearly transformed to create $Z_t, W_t$ (described in H.2) are too simple to make convincing claims about the empirical advantages of this algorithm."
>
> > **Response**: Thank you for the helpful suggestion. We fully agree that evaluating the method under richer and more challenging confounding structures can further strengthen the empirical claims. Our original confounding setup follows the design for synthetic data generation in the off-policy evaluation literature [1].
>
> >Following your feedback, we have incorporated a more complex setting, mainly following the data generation process proposed in [2]. In this simulation, each time step begins by sampling an unobserved confounder $U_t \in \mathbb{R}^{d_a} \sim \mathcal{N}(0,I)$. We then map $U_t$ through a smooth, highly nonlinear feature operator $\psi(U_t)=\frac{(U_t-0.5)^4}{40}+e^{-2.5(U_t-0.5)^2}+\frac{U_t-0.5}{6}$. The proxy $Z_t$ is generated by combining $\psi(U_t)$ with harmonics of $U_t$: $Z_t=1.4\sin(\Omega\psi(U_t)+\phi)+0.9\cos((\Omega+0.3)U_t)+0.25\psi(U_t)^2+0.1U_t^3+\varepsilon_t^{(Z)}$, where frequencies $\Omega \in [0.8,1.6]$, phases $\phi \in [0,\pi/2]$, and $\varepsilon_t^{(Z)} \sim \mathcal{N}(0,0.08^2I)$. The outcome-inducing proxy $W_t$ is generated from the same confounder via $\psi_W(U_t)=\frac{U_t^4}{55}+e^{-3U_t^2}+\frac{U_t}{5.5}$ and $W_t=0.55\psi_W(U_t)+0.2\psi_W(U_t)^2+0.85\sin(1.1U_t)+0.6\cos(0.9\psi_W(U_t))+\varepsilon_t^{(W)}$, with $\varepsilon_t^{(W)}\sim\mathcal{N}(0,0.07^2I)$. We then inject the confounder into the next states by adding the scalar $\sum_i\psi(U_{t,i})$ to every dimension of $S_{t+1}$, yielding the confounded next state. Other aspects of the experimental setup remain unchanged.
>
> >We summarize the updated experimental results in the table below. Overall, the three standard MBRL algorithms augmented with our deconfounding approach consistently achieved lower MSEs than their confounded counterpart, which do not account for unobserved confounders.
> | Methods           | Timesteps     | Swimmer       | Hopper        | Half Cheetah   | Ant            |
> |------------------|---------------|---------------|---------------|----------------|----------------|
> | **Confounded MBPO** | 1k (1-step)   | 1.73±1.93     | 3.08±2.85     | 16.56±9.52     | 38.80±25.39    |
> |                  | 10k (5-step)  | 2.35±1.77     | 3.50±3.82     | 23.96±15.93    | 43.46±20.93    |
> |                  | 20k (10-step) | 2.66±2.13     | 4.83±4.21     | 27.31±15.00    | 53.98±26.41    |
> |                  | 30k (15-step) | 2.93±1.89     | 7.72±5.46     | 43.99±23.98    | 91.06±35.81    |
> | **DPC-MBPO**     | 1k (1-step)   | 0.73±0.48     | 0.47±0.56     | 5.10±1.96      | 9.00±4.33      |
> |                  | 10k (5-step)  | 1.22±0.78     | 0.64±0.40     | 12.13±6.08     | 14.61±5.15     |
> |                  | 20k (10-step) | 1.50±0.72     | 1.27±0.46     | 14.98±6.92     | 16.86±7.67     |
> |                  | 30k (15-step) | 2.49±0.93     | 2.65±1.47     | 17.52±7.88     | 25.57±8.37     |
> | **Confounded M2AC**| 1k (1-step)  | 1.62±1.78     | 2.95±3.21     | 11.36±6.85     | 27.11±15.78    |
> |                  | 10k (5-step)  | 2.41±2.21     | 3.77±3.73     | 19.55±11.70    | 27.43±15.06    |
> |                  | 20k (10-step) | 2.30±1.95     | 3.47±3.64     | 20.17±12.50    | 25.71±16.72    |
> |                  | 30k (15-step) | 2.47±2.05     | 3.87±3.72     | 22.74±11.22    | 23.25±11.31    |
> | **DPC-M2AC**     | 1k (1-step)   | 0.40±0.30     | 0.28±0.31     | 3.31±1.29      | 6.44±2.48      |
> |                  | 10k (5-step)  | 0.53±0.43     | 0.46±0.31     | 8.04±3.84      | 8.19±2.51      |
> |                  | 20k (10-step) | 0.55±0.37     | 0.58±0.41     | 9.47±4.77      | 7.78±3.24      |
> |                  | 30k (15-step) | 0.90±0.70     | 0.85±0.55     | 11.04±5.42     | 9.08±3.10      |
> | **Confounded MACURA**|1k (1-step) | 1.78±1.51     | 2.81±3.46     | 13.90±9.94     | 26.29±18.88    |
> |                  | 10k (5-step)  | 1.93±1.81     | 3.45±3.69     | 18.96±10.41    | 27.31±17.95    |
> |                  | 20k (10-step) | 2.19±1.65     | 3.55±4.30     | 18.39±10.91    | 27.40±16.02    |
> |                  | 30k (15-step) | 2.26±2.26     | 4.91±5.00     | 18.20±9.30     | 27.34±13.25    |
> | **DPC-MACURA**   | 1k (1-step)   | 0.42±0.35     | 0.47±0.43     | 3.54±1.58      | 7.66±3.49      |
> |                  | 10k (5-step)  | 0.83±0.60     | 0.64±0.55     | 8.69±4.33      | 8.95±3.59      |
> |                  | 20k (10-step) | 0.84±0.63     | 0.96±0.77     | 9.59±4.53      | 8.70±3.59      |
> |                  | 30k (15-step) | 0.98±0.56     | 1.76±1.12     | 10.16±6.31     | 8.76±3.20      |
>
> >We have added the results to Appendix H.4.1 of the revised paper.

---

> ### Author Response · Authors · 2025-11-27
>
> #### **Weaknesses 2**: "Algorithm 1 describes a two-step process for policy learning. However, the experiments either report the error of the learned dynamics model or the OPE estimation errors, and do not present the overall performance of the learned policy resulting from completing the full procedure in Algorithm 1. While theoretically one might expect that reducing dynamics or OPE estimation errors could lead to improved policy learning, it would strengthen the work to compare the obtained policy’s value against that of the baselines or the oracle (if the oracle optimal policy is known)."
>
> > **Response**: Thank you for this insightful suggestion. We agree that reporting the overall performance of the learned policy after completing the full procedure in Algorithm 1 would strengthen the empirical section.
>
> >To address this, we conducted an additional policy-learning demo experiment in the HalfCheetah environment. We kept the original experimental setup unchanged and introduced confounding into the reward function using the following mechanism:
> $r_t^{\text{conf}} = r_t + 0.3 \sum_i U_{t,i}$
> The resulting reward curves are shown in the figure (https://imgur.com/kIx2d78). The solid lines depict the average return of five episodes over environment steps for both methods. The shaded regions around each curve represent the variability across random seeds, corresponding to ±1 standard deviation. As illustrated, DPC- MBPO is able to learn a high-performing policy despite the presence of unobserved confounding. Although the performance gap relative to the oracle optimal policy [3] remains, it achieves substantially better policy improvement compared to Confounded-MBPO, which fails to make meaningful progress throughout training. This provides preliminary evidence that completing the full proximal model-based procedure indeed improves downstream policy learning performance.
>
> >We have added the results to Appendix H.4.4 of the revised paper.
>
> #### **Weaknesses 3**: "Another key limitation of this work, as the authors also acknowledge, is its assumption about the existence of valid proxy variables that satisfy Assumption 3.2. In realistic scenarios, obtaining proxy variables $W_t, Z_t$ may not be feasible. This also speaks to my concern raised earlier that the MuJoCo environments and the data-generating process (i.e., transforming $U_t$ to $Z_t$ and $W_t$ via a tanh function and adding a very small Gaussian noise) are too simplistic to effectively model real-world systems."
>
> > **Response**: We thank the reviewer for raising this important point. We agree that in general obtaining valid auxiliary variables such as the proxies required in our paper or instrumental variables is not always straightforward in real environments. Fortunately, in many cases, domain knowledge can provide sufficient guidance on this, and some applied studies, e.g., [4-5] have drawn on domain expertise to identify auxiliary variables that meet relevant assumptions, thereby enabling deconfounding. They also provide practical, scenario-specific guidance on how to select them. Here, we provide two examples:
>
> >First, [4] shows that insurance plans can serve as an auxiliary variable for a patient’s treatment assignment, as it influences the treatment choice but is independent of the clinical unmeasured confounders. Second, [6] uses the patient’s oxygenation status as an action-inducing proxy for whether clinicians decide to perform right-heart catheterization (RHC), given its strong correlation with the intubation decision. Then hematocrit is used as an outcome-inducing proxy because it is associated with the eventual clinical outcome.
>
> >In addition, in scenarios where expertise knowledge is limited, it would be possible to learn the proxies from data directly. Learning the representations of auxiliary variables, especially instrumental variables from data is an emerging research topic in the causal inference field [7-8]. Following this line of work, our approach can also be extended by incorporating representation learning to automatically construct the proxy variables from raw observations, which will be a very promising future direction for us. We have mentioned this challenge and outlined this future research direction in the Conclusion section of the paper.

---

> ### Author Response · Authors · 2025-11-27
>
> #### **Questions 1**: "For both experiments (Table 1 and the box plot in Figure 3), it would be beneficial to present the overall performance of the learned policy obtained after completing the full procedure described in Algorithm 1. This would also allow the authors to compare their results with model-free baselines in POMDPs with unobservable confounders (e.g., Tchetgen Tchetgen et al., 2024; Miao et al., 2024), as the current Table 1 only compares against the confounded model-based RL setting ("Confounded MBPO") and lacks comparisons with other baselines that would help contextualize the strength of the proposed method."
>
> > **Response**: Thank you again for this helpful suggestion. We agree that presenting the overall performance of the learned policy would further strengthen the contribution. We have included the overall policy learning performance comparison between confounded MBPO and DPC-MBPO in the HalfCheetah environment, as demonstrated in our response to W2.
>
> >For the RHC experiment, existing model-free RL for confounded POMDPs (e.g., Tchetgen Tchetgen et al., 2024; Miao et al., 2024) primarily focus on off-policy evaluation (OPE), that is, estimating the value of a target policy from observational data without directly executing the policy in the environment. These approaches do not contain a policy optimization/learning component and therefore do not provide a mechanism to perform the two-step procedure of Algorithm 1. As a result, it may not be directly feasible to apply existing model-free methods for confounded POMDPs (e.g., Tchetgen Tchetgen et al., 2024; Miao et al., 2024) to perform policy optimization in the MuJoCo environments. At the same time, this also reveals a promising future direction, i.e., seeking optimal policies in confounded POMDPs using model-free approaches.
>
> #### **References**:
> >[1] Miao R, Qi Z, Zhang X. "Off-policy evaluation for episodic partially observable Markov decision processes under non-parametric models." NeurIPS, 2022.
> [2] Kompa B, Bellamy D, Kolokotrones T, Beam A. "Deep learning methods for proximal inference via maximum moment restriction." NeurIPS, 2022.
> [3] Janner M, Fu J, Zhang M, Levine S. "When to trust your model: Model-based policy optimization." NeurIPS, 2019.
> [4] Ertefaie A, Small DS, Flory JH, Hennessy S. "A tutorial on the use of instrumental variables in pharmacoepidemiology." Pharmacoepidemiology and Drug Safety, 2017.
> [5] Brookhart MA, Rassen JA, Schneeweiss S. "Instrumental variable methods in comparative safety and effectiveness research." Pharmacoepidemiology and Drug Safety, 2010.
> [6] Shen T, Cui Y. "Optimal treatment regimens for proximal causal learning." NeurIPS, 2023.
> [7] Cheng D, Xu Z, Li J, Liu L, Le TD, Liu J. "Learning conditional instrumental variable representation for causal effect estimation." ECML/PKDD, 2023.
> [8] Huang Z, Zhang S, Cheng D, Li J, Liu L, Lu G, Zhang G. "Learning instrumental variable representation for debiasing in recommender systems." Neural Networks, 2025.

---

### Official Review · Reviewer_Uo35 · 2025-10-31

**Soundness:** 2
**Presentation:** 2
**Contribution:** 2
**Rating:** 2
**Confidence:** 5

**Summary:**

The paper studies model-based RL for confounded POMDPs by leveraging proximal causal inference with dynamic- and reward-emission bridge functions. Building on an identification result for the policy value and conditional moment restrictions for the bridge functions, the authors propose learning the bridges with neural networks by minimizing a maximum moment restriction (MMR) objective. Numerical experiments in simulated control tasks and on a real-world clinical dataset demonstrate the method’s effectiveness.

**Strengths:**

1. Motivation is clear: The problem of confounded POMDPs is challenging, and there is limited practical work on model-based algorithms for this setting. This paper presents a practical algorithm with general function approximation using deep neural networks.

2. Empirical evidence is supportive: The proposed method outperforms baselines in confounded POMDP settings, and its application to a real-world clinical dataset is promising.

3. Presentation is well organized.

**Weaknesses:**

In general, the contributions feel limited, and several major concerns remain.

1. Originality of the theoretical result: The theoretical guarantee (presumably Theorem 3.1) is highlighted as a primary contribution. However, a similar result appears in Theorem 3.5 of [1]. The paper does not clearly articulate the novelty—either in the statement or in the proof techniques—relative to [1]. I suggest to add more discussions on the technical contributions.

2. Alignment between identification and estimation: The bridge functions used for identification (Theorem 3.1) and those used for estimation (Assumption 3.2; Section 3.2) appear misaligned. In particular, the identification bridges involve the immediate reward and the next state, whereas the estimation bridges do not. Based on lines 928–931 in the appendix, it seems you may need conditional moment restrictions that hold pointwise in the immediate reward and next state rather than only in expectation. I suggest the authors to carefully revisit this point.

3. How estimated bridges are used in policy evaluation: Given the above point#2, it is unclear how the estimated bridge functions are actually used to perform policy evaluation under Theorem 3.1. Adding step-by-step details (or pseudocode) in Algorithm 1 would help make the pipeline explicit.




[1] Model-based Reinforcement Learning for Confounded POMDPs. Hong et. al.

**Questions:**

1. My understanding is that the main results are stated for the finite-horizon setting, while Section 2.1 also introduces a discounted infinite-horizon formulation. Could you clarify it?

2. Methodological trade-offs: Could the authors provide more insight into the trade-offs between model-based approaches and existing model-free methods for confounded POMDPs [1][2]?

[1] Off-policy evaluation for episodic partially observable markov decision processes under non-parametric models. Miao et. al.

[2] Proximal reinforcement learning: Efficient off-policy evaluation in partially observed markov decision processes. Bennett et. al.

---

> ### Author Response · Authors · 2025-11-27
>
> #### **Weaknesses 1**: "Originality of the theoretical result: The theoretical guarantee (presumably Theorem 3.1) is highlighted as a primary contribution. However, a similar result appears in Theorem 3.5 of [1]. The paper does not clearly articulate the novelty—either in the statement or in the proof techniques—relative to [1]. I suggest to add more discussions on the technical contributions."
>
> >**Response**: Thank you for this insightful comment. We acknowledge that our theoretical guarantee in Theorem 3.1 may appear like the identification results in prior work [1] (i.e. Hong et al.). In fact, this type of sequential factorization of the trajectory distribution is a common and widely used paradigm in policy evaluation field. For example, the theoretical results in Theorem 4.1 of [2], Theorem 1 of [3], and Theorem 1 of [4], all follow this generic pattern, which leverages causal inference to iteratively construct consistent identification of a target policy in the presence of unobserved confounders, step by step along the trajectory.
>
> >We would like to clarify that our main contribution is not the proposal of Theorem 3.1. Instead, in contrast to prior works in RL for confounded POMDPs, our work targets two specific gaps that remain open in the existing literature:
> (1) Model-free proximal causal inference assisted RL methods [3-5] being grounded in the Bellman equation, cannot explicitly model the next-step transition dynamics, which are indispensable for planning-based approaches such as model predictive control (MPC).
> (2) Existing model-based approaches, such as [1] and [6], rely on prespecified kernel functions and tabular model, which limit their applicability in more complex environments.
>
> >Motivated by these limitations, our work develops a neural network-based proximal causal identification framework that does not rely on prespecified kernel functions or tabular model assumptions. This design aligns closely with mainstream model-based RL practices [7-8], where neural networks are used to model the transition dynamics, and therefore offers greater flexibility in complex or high-dimensional environments. Although theoretical expressions and their derivations may resemble the formulations of existing works, the modeling (neural network) and estimation (NMMR) of the deconfounded dynamics/reward model in our work differ from those used in existing works.
>
> #### **Weaknesses 2**: "Alignment between identification and estimation: The bridge functions used for identification (Theorem 3.1) and those used for estimation (Assumption 3.2; Section 3.2) appear misaligned. In particular, the identification bridges involve the immediate reward and the next state, whereas the estimation bridges do not. Based on lines 928–931 in the appendix, it seems you may need conditional moment restrictions that hold pointwise in the immediate reward and next state rather than only in expectation. I suggest the authors to carefully revisit this point."
>
> >**Response**:Thank you very much for pointing out the notational inconsistency between the bridge functions used for identification (Theorem 3.1) and those presented in Assumption 3.2 (Section 3.2). In the original submission, Theorem 3.1 employs distribution bridges, while Assumption 3.2 was stated using a mean bridge form. This mismatch in notation is the source of confusion.
>
> >To resolve this, we have revised Assumption 3.2 to the distribution bridge formulation. Specifically, we assume the existence of reward-emission bridge functions $h_R(r_t, s_t, a_t, w_t)$ and dynamic-emission bridge functions $h_D(s_{t+1}, s_t, a_t, w_t)$, such that for all $(s_t, a_t, z_t)$ at each time step $t = 1, ..., T$:
> $p(r_t \mid s_t, a_t, z_t) = \mathbb{E}[h_R(r_t, s_t, a_t, W_t) \mid s_t, a_t, z_t]$, and,
> $p(s_{t+1} \mid s_t, a_t, z_t) = \mathbb{E}[h_D(s_{t+1}, s_t, a_t, W_t) \mid s_t, a_t, z_t]$.
> >This formulation serves the same role as the bridge functions in the previous version, in the sense that, given the current state $s_t$, action $a_t$, and proxy $z_t$, the distribution of the next state/reward can be recovered by taking the conditional expectation of the bridge functions $h_D$ or $h_R$ over proxy $W_t$. The only difference is that the previous bridge functions recovered the mean of the next state/reward, whereas the current version identifies the entire distribution.
>
> >In addition, we have updated the proof of identification result in Appendix E and estimation of bridge function in Section 3.2 based on the revised assumptions. Finally, we sincerely thank the reviewer again for the careful reading and insightful comments.

---

> ### Author Response · Authors · 2025-11-27
>
> #### **Weaknesses 3**: "How estimated bridges are used in policy evaluation: Given the above point#2, it is unclear how the estimated bridge functions are actually used to perform policy evaluation under Theorem 3.1. Adding step-by-step details (or pseudocode) in Algorithm 1 would help make the pipeline explicit."
>
> >**Response**: Thank you for pointing this out. Theorem 3.1 identifies the policy value by factorizing the interventional trajectory distribution under the target policy into sequential components and then using the estimated reward and dynamics bridge functions to replace the parts of this factorization iteratively. Each factor in this sequential factorization corresponds to an outcome under an interventional action, e.g., $p_r (r_t∣s_t,do(a_t ))$ and $p_s (s_{t+1}∣s_t,do(a_t ) )$, which cannot be obtained directly from observational data with unobserved confounders.
>
> >The above identification process mirrors the way MBRL algorithms generate simulated data using the learned models (Line 6 in Algorithm 1). Under the agent’s policy π, we iteratively simulate trajectories using the dynamics and reward models identified via the estimated bridge functions. Theorem 3.1 ensures that these model components, obtained through the reward and dynamics bridges, yield unbiased predictions of the next state and reward under policy π. This point is also highlighted in Remark 1.
>
> >We have added further explanations in Section 3.3 on how estimated bridges are used in policy evaluation.
>
> #### **Questions 1**: "My understanding is that the main results are stated for the finite-horizon setting, while Section 2.1 also introduces a discounted infinite-horizon formulation. Could you clarify it?"
>
> >**Response**: Thank you for your question. The confusion arises because the background and notation in Section 2.1 follows some standard conventions in the RL literature [7-8], where the discounted return is often written in a way that resembles the infinite-horizon formulation. However, our main theoretical results are under a finite horizon setting. The $T$ that appears in Section 2.1 denotes a terminal time, and as clarified in Lemma 1, our identification result holds “over a finite horizon $T$.”
>
> #### **Questions 2**: "Methodological trade-offs: Could the authors provide more insight into the trade-offs between model-based approaches and existing model-free methods for confounded POMDPs [1][2]?"
>
> >**Response**:  Thank you for raising this point. The fundamental distinction between model-free and model-based methods lies in whether explicitly model the environment dynamics. Model-based methods learn the transition and reward models, enabling the reuse of the learned model for simulated rollouts. This typically brings higher sample efficiency. However, MBRL may suffer from model misspecification if the learned dynamics are inaccurate. In contrast, model-free methods avoid modeling the dynamics and directly estimate value functions, which reduces the risk of model bias but often requires substantially more real data and can lead to higher variance.
>
> #### **References**:
> >[1] Hong M, Qi Z, Xu Y. "Model-based reinforcement learning for confounded POMDPs." ICML, 2024.
> [2] Xu Y, Zhu J, Shi C, Luo S, Song R. "An instrumental variable approach to confounded off-policy evaluation." ICML, 2023.
> [3] Tennenholtz G, Shalit U, Mannor S. "Off-policy evaluation in partially observable environments." AAAI, 2020.
> [4] Bennett A, Kallus N. "Proximal reinforcement learning: Efficient off-policy evaluation in partially observed Markov decision processes." Operations Research, 2024.
> [5] Miao R, Qi Z, Zhang X. "Off-policy evaluation for episodic partially observable Markov decision processes under non-parametric models." NeurIPS, 2022.
> [6] Gasse M, Grasset D, Gaudron G, Oudeyer PY. "Using confounded data in latent model-based reinforcement learning." TMLR, 2023.
> [7] Frauenknecht B, Eisele A, Subhasish D, Solowjow F, Trimpe S. "Trust the Model Where It Trusts Itself — Model-Based Actor-Critic with Uncertainty-Aware Rollout Adaption." ICML, 2024.
> [8] Janner M, Fu J, Zhang M, Levine S. "When to trust your model: Model-based policy optimization." NeurIPS, 2019.

---

### Official Review · Reviewer_qSzC · 2025-11-11

**Soundness:** 2
**Presentation:** 2
**Contribution:** 1
**Rating:** 4
**Confidence:** 3

**Summary:**

This paper investigates model-based reinforcement learning (MBRL) in environments where unobserved confounders affect both actions and outcomes, a setting termed confounded partially observable Markov decision processes (confounded POMDPs). Existing methods either rely on tabular formulations or kernel-based nonparametric models, both of which are impractical for continuous and high-dimensional state spaces. To address these limitations, the authors propose a Deep Proximal Causal MBRL (DPC-MBRL) method that integrates proximal causal inference with neural network–based dynamics modeling. The paper first provides a theoretical identification result showing how the policy value can be consistently estimated under confounding using bridge functions with proxy variables. Then, it proposes a neural-network-based estimation of these bridge functions, offering a flexible and scalable approach. Empirically, DPC-MBRL is evaluated on MuJoCo control tasks and a real-world medical dataset. The experiments show that DPC-MBRL mitigates confounding bias more effectively and yields better model estimation accuracy than existing MBRL and model-free deconfounding methods.

**Strengths:**

1. The theoretical development is relatively rigorous and grounded in established causal inference literature (e.g., Miao et al., 2018; Tchetgen Tchetgen et al., 2024). The proofs (Appendix E) clearly demonstrate the unbiasedness of the identified policy value.
2. This work contributes toward causally robust model-based RL—an underexplored area compared to model-free causal RL. Given the growing relevance of confounded observational data (robotics, healthcare, offline RL), this framework could have broad implications for safe and reliable deployment of RL systems.

**Weaknesses:**

1. The approach assumes access to valid action-inducing and outcome-inducing proxies satisfying the completeness condition. While the paper provides conceptual examples and guidance (Appendix D), in real environments identifying or constructing such proxies remains challenging.
2. The overall theoretical novalty of the proposed method seems limited, given existing works on proximal causal inference assisted RL and corresponding theoretical results, including model-based methods, e.g., Hong et al, 2024.
3. From the perspective of RL, the presentation of the paper seems confusing. The initial development of the theoretical setup seems to be describing the observational data (offline data) generating process which involves confounding. But the algorithm framework (Algorithm 1) seems to be describing an online learning algorithm where the learned policy can be directly deployed and thus there should be no confounding issue. How should I understand that?

**Questions:**

1. Can the authors propose or test methods to *learn* proxy variables directly from raw observations, perhaps via auxiliary networks or latent representation learning?
2. How sensitive is the approach if the selected proxies violate the completeness assumption? Would errors propagate to the learned policy?
3. What is the empirical runtime overhead of DPC-MBRL compared to vanilla MBPO or model-free methods?

---

> ### Author Response · Authors · 2025-11-26
>
> #### We sincerely appreciate your detailed comments and your recognition of our work. We provide our responses to your questions below.
>
> #### **Weakness 1**: "The approach assumes access to valid action-inducing and outcome-inducing proxies satisfying the completeness condition. While the paper provides conceptual examples and guidance (Appendix D), in real environments identifying or constructing such proxies remains challenging."
>
> > **Response**: We thank the reviewer for raising this important point. We agree that in general obtaining valid auxiliary variables such as the proxies required in our paper or instrumental variables is not always straightforward in real environments. Fortunately, in many cases, domain knowledge can provide sufficient guidance on this, and some applied studies, e.g., [1-2] have drawn on domain expertise to identify auxiliary variables that meet relevant assumptions, thereby enabling deconfounding. They also provide practical, scenario-specific guidance on how to select them. Here, we provide two examples:
>
> >First, [1] shows that insurance plans can serve as an auxiliary variable for a patient’s treatment assignment, as it influences the treatment choice but is independent of the clinical unmeasured confounders. Second, [3] uses the patient’s oxygenation status as an action-inducing proxy for whether clinicians decide to perform right-heart catheterization (RHC), given its strong correlation with the intubation decision. Then hematocrit is used as an outcome-inducing proxy because it is associated with the eventual clinical outcome.
>
> >In addition, as the reviewer has pointed out in Q1, in scenarios where expertise knowledge is limited, it would be possible to learn the proxies from data directly. Learning the representations of auxiliary variables, especially instrumental variables from data is an emerging research topic in the causal inference field [4-5]. Following this line of work, our approach can also be extended by incorporating representation learning to automatically construct the proxy variables from raw observations, which will be a very promising future direction for us. We have mentioned this challenge and outlined this future research direction in the Conclusion section of the paper.
>
> #### **Weakness 2**: "The overall theoretical novalty of the proposed method seems limited, given existing works on proximal causal inference assisted RL and corresponding theoretical results, including model-based methods, e.g., Hong et al, 2024."
>
> > **Response**: We thank the reviewer for the comment. We agree that proximal causal inference has been applied to reinforcement learning [6-10]. A common feature of these works is that they leverage proxy variables to construct consistent identification of the target policy value. The resulting expressions for the policy value typically follow a sequential-decision structure, which leads to theoretical formulations that appear similar across different studies. For example, the theoretical results in Theorem 4.1 of [11], Theorem 1 of [12], and Theorem 1 of [13] can all be viewed as instances of this general paradigm. The primary theoretical innovations in these works therefore lie in selecting the appropriate proximal causal learning technique to address the specific type of identification gap.
>
> >Then, we would like to clarify that our main novelty. In contrast to prior works in RL for confounded POMDPs, our work targets two specific gaps that remain open in the existing literature:
> (1) Model-free proximal causal inference assisted RL methods [6-8] being grounded in the Bellman equation, cannot explicitly model the environment dynamics, which are indispensable for planning-based approaches such as model predictive control (MPC).
> (2) Existing model-based approaches, such as [9] and [10], rely on prespecified kernel functions and tabular model, which limit their applicability in more complex environments.
>
> >Motivated by these limitations, our work develops a neural network-based proximal causal identification framework that does not rely on prespecified kernel functions or tabular model assumptions. This design aligns closely with mainstream model-based RL practices [14-15], where neural networks are used to model the environment dynamics, and therefore offer greater flexibility in complex or high-dimensional environments. Although the theoretical expressions and their derivations in our work may resemble existing formulations, the modeling (neural network) and estimation (NMMR) of the deconfounded transition dynamics in our theoretical expressions differ from the prior works.
>
> >We have mentioned this in the Introduction and Appendix B of the revised paper.

---

> ### Author Response · Authors · 2025-11-26
>
> #### **Weakness 3**: “From the perspective of RL, the presentation of the paper seems confusing. The initial development of the theoretical setup seems to be describing the observational data (offline data) generating process which involves confounding. But the algorithm framework (Algorithm 1) seems to be describing an online learning algorithm where the learned policy can be directly deployed and thus there should be no confounding issue. How should I understand that?”
>
> > **Response**: We thank the reviewer for the question. First, Algorithm 1 is a standard model-based online RL algorithm (akin to Dyna-style MBRL) [14-15]. In Algorithm 1, Line 3 collects trajectory data under the policy π from the real environment. Since these trajectory data are generated in an environment with unobserved confounders, the dynamics model trained in Line 4 is inevitably biased, which in turn biases the simulated trajectories τ ̃ produced in Line 6. This is why deconfounding is required and where our theoretical framework plays its role.
>
> >By leveraging proxy variables and bridge functions, our method recovers unbiased dynamics/reward models from confounded data (Line 4). Given such an unbiased dynamics/reward model, the unbiased simulated trajectories can be generated in Line 6, which include consistent estimates of the next state and reward at each time step. Our theoretical setup (Theorem 3.1) guarantees that, under the policy π, the next state and reward obtained from rolling out the learned model are consistent estimates. Since the operations in Lines 6–8 do not introduce new sources of confounding, no further deconfounding procedures are necessary.
>
> >Although learning dynamics and reward models use trajectory data collected from a true environment under policy π, making this stage appear offline, the procedure is inherently part of the standard online RL learning process.
>
> #### **Questions 1**: “Can the authors propose or test methods to learn proxy variables directly from raw observations, perhaps via auxiliary networks or latent representation learning?"
>
> > **Response**:  Thank you for the suggestion. We agree that this is an important and promising direction, and it aligns well with our future work in the Conclusion section. As mentioned in our response to W1, in scenarios where expertise knowledge is limited, it would be possible to learn the proxies from data directly. Learning the representations of auxiliary variables, especially instrumental variables from data is an emerging research topic in the causal inference field [4-5]. Following this line of work, our approach can also be extended by incorporating representation learning to automatically construct the proxy variables from raw observations, which will be a very promising future direction for us.
>
> #### **Questions 2**: “How sensitive is the approach if the selected proxies violate the completeness assumption? Would errors propagate to the learned policy?"
>
> > **Response**: Thank you for the insightful question. We acknowledge that our approach relies on the completeness assumption for the correct estimation of the bridge functions. Following your comment, to explicitly illustrate the importance of this assumption, we have conducted a sensitivity analysis where we intentionally violate the completeness assumption by perturbing the proxy $W_t$. Specifically, we keep the generation of the unobserved confounder $U_t$ and the proxy $Z_t$ as previously described in the paper, but replace the outcome proxy $W_t$ with a nearly constant signal: $W_t = c + \varepsilon_t,\quad \varepsilon_t \sim \mathcal{N}(0,\sigma^2 I),$ where $c$ is a fixed vector (each entry of it is set to 0.1) and $\sigma=0.05$. Because such a nearly constant $W_t$ carries almost no information about the unobserved confounder, thereby violating the completeness assumption [16]. The results are summarized in the following table:
> | Methods      | Time steps  | Swimmer     | Hopper      | Half Cheetah | Ant        |
> |--------------|------------------------------|-------------|-------------|---------------|------------|
> | **DPC-MBPO** | 1k (1-step)                  | 0.64±0.63   | 1.05±1.04   | 3.01±2.83     | 8.42±5.09  |
> |              | 10k (5-step)                 | 0.69±0.69   | 0.95±0.82   | 8.31±4.4      | 9.26±4.57  |
> |              | 20k (10-step)                | 0.85±0.67   | 1.85±1.89   | 10.18±4.96    | 9.68±4.17  |
> |              | 30k (15-step)                | 1.17±0.97   | 2.37±1.33   | 11.15±5.14    | 11.98±4.38 |
> | **DPC-M2AC** | 1k (1-step)                  | 0.61±0.58   | 0.89±1.11   | 2.54±2.68     | 8.97±6.50  |
> |              | 10k (5-step)                 | 0.79±0.73   | 1.06±0.97   | 7.38±4.80     | 8.49±4.23  |
> |              | 20k (10-step)                | 0.63±0.64   | 1.72±1.47   | 8.49±4.46     | 9.45±5.24  |
> |              | 30k (15-step)                | 0.75±0.72   | 1.60±1.38   | 8.97±4.72     | 8.77±5.80  |

---

> ### Author Response · Authors · 2025-11-26
>
> >Regarding Question 2 — continued table is provided below:
> | Methods        | Timesteps     | Swimmer   | Hopper    | Half Cheetah | Ant       |
> | -------------- | ------------- | --------- | --------- | ------------ | --------- |
> | **DPC-MACURA** | 1k (1-step)   | 0.71±0.69 | 1.11±1.23 | 2.68±2.83    | 8.19±5.55 |
> |                | 10k (5-step)  | 0.81±0.71 | 1.10±1.14 | 6.80±3.78    | 8.32±5.73 |
> |                | 20k (10-step) | 0.72±0.64 | 1.41±1.40 | 9.37±5.49    | 7.45±4.53 |
> |                | 30k (15-step) | 0.71±0.58 | 1.49±1.32 | 9.60±7.59    | 7.27±3.35 |
>
> >Across all environments, we observe that once completeness is violated, the learned bridge function becomes biased, leading to higher model estimation errors (larger MSE) compared with the original setting where the assumption holds (See Table 1 in the paper). This systematically demonstrates that completeness is crucial for accurate identification.
>
> >We have added the results to Appendix H.4.3 of the revised paper.
>
>
> #### **Questions 3**: “What is the empirical runtime overhead of DPC-MBRL compared to vanilla MBPO or model-free methods?"
>
> > **Response**: Thanks for this question. Our experiments were conducted using the National Computational Infrastructure (NCI Australia), and all methods were run under the same hardware configuration aligned with the default compute resource specification for general-user accounts. The runtime information is as follows:
> | Method              | Runtime     |
> | ------------------- | ----------- |
> | MBPO (vanilla)      | ~10 minutes |
> | pFOE                | ~40 minutes |
> | PCI                 | ~1 hour     |
> | DPC-MBRL (ours) | ~1.2 hours  |
>
> >The additional runtime of ours arises from the proximal causal components, which require computing kernel-based moment conditions. In practice, this involves a Gaussian kernel using pairwise distance evaluations and combining multiple kernel matrices. These steps introduce inherent O(n²)–O(n³) computational and memory costs, making our method slower than purely feed-forward baselines like vanilla MBPO. In addition, the model-free baselines pFQE and PCI also require comparatively more training time. Both methods rely on components such as doubly robust estimators, alternating updates between value and policy objectives, and regularization terms that must be optimized jointly, which naturally increase their overall wall-clock time.
>
> >We have added the results to Appendix I.3 of the revised paper.
>
> #### **Reference**:
> >[1] Ertefaie A, Small DS, Flory JH, Hennessy S. "A tutorial on the use of instrumental variables in pharmacoepidemiology." Pharmacoepidemiology and Drug Safety, 2017.
> [2] Brookhart MA, Rassen JA, Schneeweiss S. "Instrumental variable methods in comparative safety and effectiveness research." Pharmacoepidemiology and Drug Safety, 2010.
> [3] Shen T, Cui Y. "Optimal treatment regimes for proximal causal learning." NeurIPS, 2023.
> [4] Cheng D, Xu Z, Li J, Liu L, Le TD, Liu J. "Learning conditional instrumental variable representation for causal effect estimation." ECML/PKDD, 2023.
> [5] Huang Z, Zhang S, Cheng D, Li J, Liu L, Lu G, Zhang G. "Learning instrumental variable representation for debiasing in recommender systems." Neural Networks, 2025.
> [6] Tennenholtz G, Shalit U, Mannor S. "Off-policy evaluation in partially observable environments." AAAI, 2020.
> [7] Miao R, Qi Z, Zhang X. "Off-policy evaluation for episodic partially observable Markov decision processes under non-parametric models." NeurIPS, 2022.
> [8] Bennett A, Kallus N. "Proximal reinforcement learning: Efficient off-policy evaluation in partially observed markov decision processes." Operations Research, 2024.
> [9] Hong M, Qi Z, Xu Y. "Model-based reinforcement learning for confounded POMDPs." ICML, 2024.
> [10] Gasse M, Grasset D, Gaudron G, Oudeyer PY. "Using confounded data in latent model-based reinforcement learning." TMLR, 2023.
> [11] Xu Y, Zhu J, Shi C, Luo S, Song R. "An instrumental variable approach to confounded off-policy evaluation." ICML, 2023.
> [12] Tennenholtz G, Shalit U, Mannor S. "Off-policy evaluation in partially observable environments." AAAI, 2020.
> [13] Bennett A, Kallus N. "Proximal reinforcement learning: Efficient off-policy evaluation in partially observed markov decision processes." Operations Research, 2024.
> [14] Kurutach T, Clavera I, Duan Y, Tamar A, Abbeel P. "Model-ensemble trust-region policy optimization." ICLR, 2018.
> [15] Yu T, Thomas G, Yu L, Ermon S, Zou JY, Levine S, Finn C, Ma T. "MOPO: Model-based offline policy optimization." NeurIPS, 2020.
> [16] Huang M, McCartan C. "Relative bias under imperfect identification in observational causal inference." arXiv, 2025.

---

### Meta-Review · Area_Chair_ck6t · 2025-12-29

**Summary:**

This paper studies model-based reinforcement learning under unobserved confounding, which is an important problem, especially under POMDPs, but the reviewers have multiple important concerns that collectively led to a marginal or negative recommendation for this paper.
While one reviewer (dwrX) was slightly positive, describing the paper as marginally above the acceptance bar, the majority judged it as below the bar due to theoretical overlap, restrictive assumptions, and limited empirical persuasiveness.

To improve the paper, I recommend that the authors provide stronger theoretical justification, clearer algorithmic presentation, and more convincing empirical evidence to reach a publishable standard.

**Reviewer Concerns:**

Concerns Addressed by the Rebuttal

- Reviewers Uo35 and qSzC noted confusion in the theoretical setup. The authors responded by revising the theoretical notation (rewriting Assumption 3.2 in its distribution-bridge form), updating the Appendix proofs, and clarifying that Algorithm 1.

- Reviewers dwrX and qSzC criticized the initial experiments for measuring only model or OPE errors rather than complete policy-learning performance. The authors added a policy-learning demonstration in the HalfCheetah environment, showing clear performance gains for DPC-MBPO compared to a confounded baseline.

- Reviewers ago2  and qSzC highlighted the lack of explanation about how proxy variables were chosen in the SUPPORT dataset, and questioned the method’s robustness when proximal assumptions fail.  The rebuttal provided a detailed account of proxy selection and appended new experiments in which completeness or exclusion assumptions were intentionally violated.

- Reviewer Uo35 asked for clearer discussion contrasting model-based versus model-free approaches and for clarification about finite- vs. infinite-horizon assumptions. The authors explicitly differentiated between these methodological regimes and confirmed that their theoretical results hold under a finite-horizon setting. This clarification adequately resolves the reviewer’s questions.

Concerns Partially Addressed or Still Outstanding

- Reviewers Uo35 and ago2 noted that the paper’s core theoretical contribution closely mirrors prior results (e.g., Hong et al., 2024; Bennett & Kallus 2024). The rebuttal clarifies positioning but does not introduce substantially new theoretical insights. The main novelty therefore remains incremental rather than conceptual.

- Reviewer ago2 explicitly requested comparisons with “Delphic Offline RL” (Pace et al., 2023), given its relevance to confounded data settings. The updated experiments improve credibility but still do not comprehensively demonstrate superiority across varied or challenging settings.

- Reviewers ago2 and qSzC stated that the method depends on some unverifiable assumptions. The authors offered domain examples and future directions via representation learning without practical demonstrations.

- Reviewer ago2 noted that the section on bridge-function estimation primarily applies the existing NMMR framework (Kompa et al., 2022) without methodological innovation. This issue remains unresolved. The contribution is positioned but still lacks methodological advancement beyond NMMR.

**Reviewer Scores:**

Overall, most reviewers would slightly improve their scores after the rebuttal.  Reviewers qSzC and ago2 would probably move from below-threshold to borderline, acknowledging clearer exposition and stronger experiments but still doubting novelty. Reviewers dwrX  might not likely upgrade the rating given the raised problem is partically resolved.

---

### Decision · Program_Chairs · 2026-01-26

Reject